# Revealing the sulfur dioxide emission reductions in China by assimilating surface observations in WRF-Chem

Tie Dai[1, 2*], Yueming Cheng[1, 2], Daisuke Goto[3], Yingruo Li[4], Xiao Tang[5], Guangyu Shi[1, 2], and Teruyuki Nakajima[3]

[1]State Key Laboratory of Numerical Modeling for Atmospheric Sciences and Geophysical Fluid Dynamics, Institute of Atmospheric Physics, Chinese Academy of Sciences, Beijing, China.

[2]Collaborative Innovation Center on Forecast and Evaluation of Meteorological Disasters, Nanjing University of Information Science and Technology, Nanjing, China.

[3]National Institute for Environmental Studies, Tsukuba, Japan.

[4]Environmental Meteorology Forecast Center of Beijing-Tianjin-Hebei, China Meteorological Administration, Beijing, China

[5]State Key Laboratory of Atmospheric Boundary Layer Physics and Atmospheric Chemistry, Institute of Atmospheric Physics, Chinese Academy of Sciences, Beijing, China

*Correspondence to*: Tie Dai (daitie@mail.iap.ac.cn)

**Abstract.** The anthropogenic emission of the sulfur dioxide ($SO_2$) over China has significantly declined as the consequence of clean air actions. In this study, we have developed a new emission inversion system based on a Four-Dimensional Local Ensemble Transform Kalman Filter (4D-LETKF) and the Weather Research and Forecasting model coupled with Chemistry (WRF-Chem) to dynamically update the $SO_2$ emission grid by grid over China by assimilating the ground-based hourly $SO_2$ observations. Sensitivity tests for the assimilation system have been conducted firstly to tune four system parameters: ensemble size, horizontal and temporal localization lengths, and perturbation size. Our results reveal that the same random perturbation factors used throughout the whole model grids with assimilating observations within about 180 km can efficiently optimize the $SO_2$ emission, whereas the ensemble size has only little effect. The temporal localization by assimilating only the subsequent hourly observations can reveal the diurnal variation of the $SO_2$ emission, which is better than that to update the the magnitude of $SO_2$ emission every 12 hours by assimilating all the observations within the 12-hour window. The inverted $SO_2$ emission over China in November 2016 has declined by an average of 49.4% since 2010, which is well in agreement with the "bottom-up" estimation of 48.0%. Larger reductions of $SO_2$ emission are found over the priori higher source regions such as the Yangtze River Delta (YRD). The simulated $SO_2$ surface mass concentrations using two distinguished chemical reaction mechanisms are both much more comparable to the observations with the newly inverted $SO_2$ emission than those with the priori emission. These indicate that the newly developed emission inversion system can

efficiently update the SO₂ emissions based on the routine surface SO₂ observations. The reduced SO₂ emission induces the sulfate and PM$_{2.5}$ surface concentrations to decrease up to 10 $\mu g\ m^{-3}$ over the center China.

## 1 Introduction

China and India are the top two emitters of the anthropogenic sulfur dioxide (SO₂) in the world (Li et al., 2017a). SO₂ is a toxic air pollutant and the precursor of sulfate aerosol, leading to the acidification of the atmosphere and the current heavy haze problem in China (Wang et al., 2016a;Huang et al., 2014;Yao et al., 2018). Sulfate aerosol can further perturb the radiative energy budget on Earth through directly scattering solar radiation (Goto et al., 2011) and hydrological cycle by aerosol-cloud interactions (Ramanathan et al., 2001;Sato et al., 2018;Rosenfeld et al., 2019). Sulfate coating on dust leads to

a shorter lifetime of dust by increasing the deliquescence of the mixed dust, inducing a great impact on radiative properties and climate modelling (Zhang et al., 2003;Bauer et al., 2007;Fu et al., 2009;Wang et al., 2013;Qi et al., 2013;Penner, 2019). Hydrophilic polluted continental aerosols such as sulfate and mixed dust serve as cloud condensation nuclei (CCN) and thus have a substantial effect on cloud properties and the initiation of precipitation (Rosenfeld et al., 2008). The liquid and ice water paths of dust-contaminated clouds were found obviously smaller than those of dust-free conditions over Eastern Asia

(Huang et al., 2006b;Huang et al., 2006a). Asian dust altering cloud microphysics and precipitation was revealed by observations and model simulations (Liu et al., 2020;Liu et al., 2019b;Liu et al., 2019a). This, in turn, plays a key role in the climate system. To mitigate climate change and control air quality, the emission control policies especially for SO₂ implemented by China since 2006 cover all the major source sectors and have become increasingly stringent over time (Zhang et al., 2012). Consequently, the decreasing trends of SO₂ loading over China have been revealed by satellite

observations, demonstrating SO₂ emissions in China have declined by 75% during 2007-2016 (Wang et al., 2018;Li et al., 2017a). The relative change rate of SO₂ emission in China during 2010-2017 is also estimated as -62% by using the bottom-up emission inventory (Zheng et al., 2018).

The timely precise emission inventories such as SO₂ are the primary inputs to models for air quality prediction and mitigation. All the atmospheric chemistry and aerosol models rely on their descriptions of the emissions virtually, which are

mostly from the "bottom-up" emission inventories. The "bottom-up" emission inventories are compiled based on indirect information such as activity data and emission factors (Zhang et al., 2009;Kurokawa et al., 2013;Zheng et al., 2018). Due to the uncertainties of the activity rates and emission factors, large discrepancies of global and regional emissions are identified among different emission inventories (Li et al., 2018;Granier et al., 2011). It demonstrates that there is still no consensus on the best estimates for the emissions of atmospheric compounds. Moreover, the "bottom-up" anthropogenic emission

inventories often lag several years behind the present and may quickly become outdated (Zheng et al., 2018), leaving the model without up-to-date emission inventories.

The emission inversion approach can feed historical and near-real-time observations into the models, providing a top-down approach to estimate and timely update the primary emissions of air pollutants (Streets et al., 2013). Generally speaking, variational and ensemble data assimilation approaches are the two widely used methodologies to estimate the emission

fluxes of gases (such as NOx, CO, VOCs) (Tang et al., 2011;Qu et al., 2017;Wu et al., 2020;Miyazaki et al., 2012b;Cheng et al., 2010;Feng et al., 2020a) and/or aerosols (Dai et al., 2019a;Cohen and Wang, 2014;Peng et al., 2017;Yumimoto et al., 2008). The NOx emission changes over China during the COVID-19 epidemic were inferred from surface $NO_2$ observations based on ensemble data assimilation approach (Feng et al., 2020b). The emission reductions during the 2015 China Victory Day Parade were successfully detected with an ensemble data assimilation system (Chu et al., 2018). The $SO_2$ emission

inventories over China were updated on monthly or seasonal time scales assuming a linear relationship between $SO_2$ emissions and satellite observed $SO_2$ column amounts (Koukouli et al., 2018;Lee et al., 2011), known as the mass balance approach (Martin, 2003), although the sulfur chemistry especially in polluted areas as well as by the interactions of clouds should be nonlinear (Goto et al., 2011;Liao et al., 2003). Fioletov et al. (2015) described a new mass balance approach to simultaneously estimate the $SO_2$ lifetimes and emissions from large $SO_2$ point sources using satellite measurements. Based

on the variational data assimilation approach in the framework of the GEOS-Chem adjoint model, Wang et al. (2016b) developed a new sophisticated inverse modeling (IM) method to timely update monthly anthropogenic $SO_2$ emissions by assimilating the Ozone Monitoring Instrument (OMI) $SO_2$ satellite measurements. The nonlinear full sulfur chemistry and lifecycle in the atmosphere were accounted for the first time to conduct the top-down estimation of the anthropogenic $SO_2$ emissions from the GEOS-Chem adjoint model (Wang et al., 2016b). However, the great limitation to the application of

variational data assimilation approach is the requirement of developing the complicated adjoint model (Henze et al., 2007 Liang et al., 2020). The ensemble data assimilation approach requires neither linearization of the observation operator and nor an adjoint model, therefore it is much more easily implemented and flexible (Evensen, 2003). Additionally, the ensemble data assimilation and the variational data assimilation use the flow-dependent and pre-calculated model error covariances respectively (Descombes et al., 2015;Zang et al., 2016). Based on the Ensemble Square Root Filter (EnSRF) approach (Chen

et al., 2019a), the recent $SO_2$ emission changes from the year 2010 in China were successfully updated to improve the model forecast skill. An ensemble Kalman filter data assimilation system was developed to simultaneously optimize the chemical initial conditions and emissions including $SO_2$ with multi-species chemical observations (Peng et al., 2018). The effects of meteorological assimilation on $SO_2$ emission inversions were also studied recently (Peng et al., 2020).

Retrievals of $SO_2$ from satellite-based spectrometers are often contaminated by factors such as interference between ozone

and $SO_2$, and there are significant regional differences between different satellite instruments (Fioletov et al., 2013). This subsequently induces the inconsistency of the inversed regional emissions by assimilating different satellite observations (Lee et al., 2011). Meanwhile, satellite observations are usually assimilated on the monthly time scale due to data availability. Compared with satellite observations, the surface $SO_2$ observations have higher accuracy and temporal frequency. Therefore, the assimilation of intensive direct surface $SO_2$ observations can provide more spatial-temporal characteristics of emission

(Chen et al., 2019a). The China National Environmental Monitoring Centre (CNEMC) started to monitor hourly concentrations of $PM_{2.5}$ (particulate matter with diameter ≤ 2.5 micrometers), $PM_{10}$, $SO_2$, $NO_2$, CO and $O_3$ since 2012, and it had included 1436 monitoring sites from 369 cities by March 2017 (Wu et al., 2018). Those important direct intensive surface $SO_2$ observations provide a new chance to estimate the more spatial-temporal characteristics of the $SO_2$ emission in

China (Chen et al., 2019a).

Due to the limited ensemble members, the Ensemble Kalman filter (EnKF) generally has a spurious long distance correlation problem (Houtekamer and Mitchell, 2001;Miyazaki et al., 2012a). Compared with the EnKF, the Local Ensemble Transform Kalman Filter (LETKF) can assimilate measurements simultaneously over different model grids in the parallel architecture (Miyoshi et al., 2007;Hunt et al., 2007). Generally speaking, the LETKF computational time is robust with increasing observations, while that of most other ensemble Kalman filter is essentially proportional to the number of observations

(Miyoshi et al., 2007). Moreover, the global analysis is linear combinations of the ensemble members in different regions, which is not confined to the limited ensemble members and provides better results in many cases (Ott et al., 2004). A Four-Dimensional LETKF (4D-LETKF) was recently developed to assimilate hourly aerosol optical properties observed by satellite, which can avoid frequent switching between the assimilation and the ensemble aerosol forecasting to significantly reduce computational load (Dai et al., 2019b). In current study, we implement a 4D-LETKF in the Weather Research and

Forecasting model coupled with Chemistry (WRF-Chem). Our major objectives are to investigate whether 4D-LETKF together with the intensive CNEMC $SO_2$ observations can be applied to quantitatively estimate the spatially resolved changes of $SO_2$ emissions in China and how sensitive are the estimated $SO_2$ emissions to the system parameters of the 4D-LETKF.

The reminder of the paper is organized as follows. In section 2, the methodology of our emission inversion system is

described in detail. Sect. 3 presents our experimental designs and purposes. The emission inversion results and validations are provided in Sect. 4 before concluding in Sect. 5.

## 2 Methodology

In order to optimize the $SO_2$ emissions in this study, we need to formally minimize a scalar cost function $J$ in a Bayesian

framework (Hunt et al., 2007;Huneeus et al., 2012). $J$ can be formulated as the sum of the departures of a potential gridded $SO_2$ emissions x and the corresponding simulated $SO_2$ surface mass concentrations to the a priori $SO_2$ emissions $x^f$ and the CNEMC observed surface $SO_2$ concentrations $y^o$:

$$J(x) = 1/2\,(x - x^f)^T B^{-1}(x - x^f) + 1/2\,(H(x) - y^o)^T R^{-1}(H(x) - y^o) \quad (1)$$

where $H$ is the observation operator that forward the $SO_2$ emissions to the simulated CNEMC measurements; $B$ and $R$ are

the covariance matrix of the error statistics of the a priori $SO_2$ emissions and CNEMC observations.

### 2.1 Forward model and observation operator

The relationship between the emission and the surface concentration of short-lived reactive gas $SO_2$ is mainly determined by the atmospheric chemical reactions, transport and deposition. The fully coupled "online" Weather Research and Forecasting model coupled with Chemistry (WRF-Chem) version 4.1.2 (Grell et al., 2005) is served as the forward model to relate the

$SO_2$ emissions to the simulated observations of surface mass concentration in current study, which can reflect the complex nonlinear relationship between atmospheric chemical concentrations and emissions. Our primary aim is to understand how sensitive are the estimated $SO_2$ emissions to the parameters of the assimilation system, which requires huge computing

resources for sensitivity experiments as described later. Therefore, the model is configured with a domain covering most of China as shown in Fig. 1 with a relatively low horizontal resolution of 50 km and 32 vertical levels (Snyder et al., 2015). A

state-of-the-art and highly nonlinear gas phase chemical mechanism named the second generation Regional Acid Deposition Model (RADM2) (Stockwell et al., 1990) coupled with the Goddard Global Ozone Chemistry Aerosol Radiation and Transport aerosol model (Chin et al., 2000;Chin et al., 2002) (i.e., chem_opt = 303) is adopted to simulate the atmospheric sulfur cycle. The RRTMG radiation scheme with prognostic aerosols is selected to consider the aerosol direct effect on atmospheric radiation and photolysis calculations (Iacono et al., 2008). The other main selected physics are identical to those

of Dai et al. (2019a). The initial and lateral boundary meteorological conditions are from the NCEP Final (FNL) Analysis. To reduce the uncertainties associated with the meteorological fields and facilitate a more straightforward comparison of simulations and observations, the predicted wind (u, v), temperature (t), and specific humidity (q) by WRF dynamical core are also nudged to the NCEP FNL analysis every 6 hours (Dai et al., 2018). The meteorological fields in the Planetary Boundary layer (PBL) are not nudged. The WRF-Chem simulated surface gridded $SO_2$ volume mixing ratios in the unit of

parts per million ($ppmv$) are firstly converted to micrograms per cubic meter ($\mu g/m^3$) for comparing to the observations (Chen et al., 2019a) and then linearly interpolated to the CNEMC site locations.

## 2.2 SO₂ observations and uncertainties of CNEMC

The quality-assured and controlled measurements of hourly $SO_2$ surface mass concentration from the CNEMC, which is partly purposefully built for assimilation (Wu et al., 2018), are used to minimize the cost function $J$. There are a total of 1424

sites in November 2016, and those sites span most of central and eastern China and primarily locate in urban and suburban areas (Peng et al., 2017). Due to unresolved emission variations between urban and suburban, model may have large representativeness errors. To overcome the spatial scale gaps and to produce more representative observations, the super-observation is adopted to average all observations located within a model grid cell (Miyazaki et al., 2012a). Altogether 463 of 7221 model grid cells are covered by the super-observations (Fig. 1). The locations of the super-observations are assumed

as the locations of the covered model grid cells. To independently verify the assimilation results, we further randomly eliminate the super-observations located in 155 of the 463 grid cells to be assimilated. In other words, the assimilated and independent verification observation sites are randomly decided. The observation error covariance matrix $R$ is assumed diagonal, in other word, the observational error covariance is assumed uncorrelated. The observation error of CNEMC is calculated as same as Chen et al. (2019a), which contains both the measurement and representativeness errors. In the

assimilation data quality control process, $SO_2$ observation leading to absolute innovation exceeding three times of the prior total spread is considered as an outlier and discarded. The innovation is calculated as observation minus the model simulated ensemble mean observation determined from the first guess filed, and the prior total spread is the square root of the sum of the background ensemble variance and the observational error variance (Chen et al., 2019a;Rubin et al., 2016).

## 2.3 4D-LETKF

The 4D-LETKF assimilation approach generalizes a flow-dependent $B$ from ensemble simulation and finds the minimum of the cost function $J$ as following five formulas (Cheng et al., 2019):

$$\bar{x}^a = \bar{x}^f + X^f \bar{w}^a \tag{2}$$

$$\bar{w}^a = \tilde{P}^a (Y^f)^T R^{-1} f(r)(y^o - \bar{y}^f) \tag{3}$$

$$\tilde{P}^a = [(k-1)I/\rho + (Y^f)^T R^{-1} f(r) Y^f]^{-1} \tag{4}$$

$$X^a = X^f W^a \tag{5}$$

$$W^a = [(k-1)\tilde{P}^a]^{1/2} \tag{6}$$

where $\bar{x}^f$ and $\bar{x}^a$ represent the ensemble mean of the first guess (*a priori*) and analysis (*a posteriori*) SO$_2$ emissions in this study; the ensemble perturbation matrix $X$ is calculated as $x(i) - \bar{x}, \{i = 1, 2, ..., k\}$, which $k$ represents the ensemble size; the matrix $\bar{w}^a$ is the Kalman gain, which specifies the increment between the first guess and the analysis; the vector $\bar{y}^f$ represents the first guess SO$_2$ surface concentrations averaged over the ensemble members; the matrix $Y^f$ is calculated as $y^f(i) - \bar{y}^f, \{i = 1, 2, ..., k\}$; $I$ represents the identity matrix. The ensemble analyses are calculated as the sum of the $\bar{x}^a$ and each of the columns of $X^a$, which is serving as part of a priori emission information for the next analysis as described later. The multiplicative inflation factor $\rho$ is used to avoid the filter divergence, which is fixed at 1.1 to inflate the analysis covariance as same as our previous studies (Dai et al., 2019b; Cheng et al., 2019). In our implementation of the 4D-LETKF, the temporal and spatial localizations are achieved by multiplying the $R^{-1}$ by a factor $f(r)$ as described in section 3, which makes the effect of an observation on the analysis decays smoothly to zero as the time and physical distance of the observation from the analysis grid point increases (Hunt et al., 2007).

As shown in Fig. 2, each assimilation cycle with 4D-LETKF includes two steps: a first guess and a state analysis. In our implementation, the first guess is the WRF-Chem ensemble forecasting for 12 hours with hourly model output. The state analysis optimizes the SO$_2$ emissions in the past 12 hours. The advantages of 4D-LETKF used here are threefold: (1) each member of the ensemble WRF-Chem simulations is continuously integrated for 12 hours, therefore, this avoids frequent switching between the ensemble WRF-Chem forecasts and the assimilation (Peng et al., 2017;Chen et al., 2019a); (2) the asynchronous observations can be assimilated to the optimize the current state (Hunt et al., 2007;Dai et al., 2019b); (3) the assimilation window time of 12 hours could avoid filter convergence and divergence by finite ensemble samples, since more frequent assimilation forces the experiments more closer, inducing the underestimation of the model spread and overconfidence in the first guess state estimate (Schutgens et al., 2010;Miyazaki et al., 2012a;Hunt et al., 2007).

**2.4 State variable and forecast model for emission**

In this study, the state variable to be optimized is the SO$_2$ emission. A forecast model for emission is required to propagate observation information and determine the first guest for the next assimilation cycle (Miyazaki et al., 2012a). We adopt the same forecast model for SO$_2$ emission proposed by Chen et al. (2019a). The forecast model for SO$_2$ emission weights 75% and 25% toward the SO$_2$ emission ensemble $E_{t_n}^a$ from the previous analysis and the static initial prior ensemble $E_{t0}$ as following formula:

$$E_{t_{n+1}}^f = 0.75 \times M E_{t_n}^a M^T + 0.25 \times E_{t0} \tag{7}$$

where M is the identity matrix. The optimized SO$_2$ emission ensemble $E_{t_n}^a$ has SO$_2$ emissions at 12 hourly timeslots, which

are used to calculate the first guess SO₂ emission ensemble $E_{t_{n+1}}^f$ in sequence for the next assimilation cycle. The SO₂ emission inversion depends on the forecast model, therefore, sensitivity experiments for various different emission forecasts are conducted to tune the assimilation system as given in Table 1. The detail settings of the sensitivity experiments will be described in next section. As shown in Figs. S1 and S2 in the Supplement, the temporal and spatial distributions of the ensemble spread of the forecast emissions $E_{t_{n+1}}^f$ are significantly sensitive to the assimilation system parameters. The initial

prior ensemble of SO₂ emission is generated by perturbing the freely public available MIX Asian inventory $S$ for November 2010 (Li et al., 2017b). For example, the SO₂ emission for ensemble member $i$ at a given location $(x, y)$ is calculated as $f_i(x, y)S(x, y)$ (Rubin et al., 2016), and the perturbation $f_i(x, y)$, $\{i = 1, 2, \dots, k\}$, follows a lognormal distribution in the $k$-dimensional space. The mean and the variance of the perturbations $f(x, y)$ are equal to 1 and the MIX SO₂ uncertainty (i.e., 35%) (Li et al., 2017b). The horizontal perfect correlated and random uncorrelated perturbations are both created to generate

the initial prior ensemble $E_{t0}$ and the associated first guess SO₂ emission ensemble $E_{t_{n+1}}^f$ as described later. The spatial distribution of the ensemble spread of the $E_{t0}$ with either horizontal perfect correlated or random uncorrelated perturbations has the similar pattern as the MIX Asian inventory $S$, which is generally equal to 35% multiplying $S$. In MIX inventory, anthropogenic emissions are aggregated into five sectors: power, industry, residential, transportation, and agriculture. However, only the combined total emission is used in the model and updated in the analysis. It aims to decrease the degree

of freedom in the analysis (Miyazaki et al., 2012a). Ten chemical species including both gaseous and aerosol species are included in MIX inventory (Li et al., 2017b). The original monthly MIX anthropogenic emissions with a horizontal resolution of 0.25°×0.25° are remapped to the model resolution of 50 km. The residential, transportation, and agriculture emissions are allocated in the lowest model layer, whereas the power and industry emissions are allocated in the lowest seven model layers with the vertical profiles of the emission factors from the Model Inter-Comparison Study for Asia

(MICS-Asia) phase III (Chen et al., 2019b). An improved speciation framework for mapping Asian anthropogenic emissions of non-methane volatile organic compounds (NMVOC) to multiple chemical mechanisms (Li et al., 2014), is adopted to prepare the initial hourly anthropogenic emissions every 12 hours with two separated emission files (i.e., io_style_emissions = 2). We do not apply any diurnal variation for the MIX emissions. Therefore, the initial priori emissions are identical throughout the 24 hours. The emissions of aerosol species for WRF-Chem are prepared according to the study of Peng et al.

(2017). Notably, only the SO₂ emission is perturbed and optimized by CNEMC SO₂ observations in this study.

    The chemical initial conditions (i.e., atmospheric SO₂ concentrations) for the next forward simulation of the WRF-Chem ensemble are also needed to be updated with the optimal emission ensemble from the previous analysis (Peng et al., 2015;Peters et al., 2005), and this is achieved by recalculation of the WRF-Chem ensemble with the optimized emissions (Fig. 2). In other word, the WRF-Chem ensemble is performed twice in one assimilation cycle. Theoretically, the

uncertainties of the forecast SO₂ concentrations by recalculation of the WRF-Chem ensemble are dependent on the optimized emissions. Lower uncertainties of the initial SO₂ conditions for the next assimilation cycle should be found with higher accurate optimized SO₂ emissions, which in turn makes the SO₂ emission inversion more reasonable. Sensitivity

experiments for the SO₂ emission inversions as described in next section are performed to choose the best assimilation system parameters.

## 3 Experimental design

The effectiveness of 4D-LETKF is highly dependent on having sufficient spread in the ensemble members in order for the observations to impact the first guess (Rubin et al., 2016;Dai et al., 2019b;Hunt et al., 2007). The ensembles represent the uncertainty in the model first guess, therefore, the method for generating the ensemble is an important consideration for an optimal "top-down" emission inversion. Meanwhile, 4D-LETKF allows a flexible choice of observations to be assimilated for a specific grid point through horizontal, vertical, and temporal observation localizations (Miyoshi et al., 2007;Dai et al., 2019b;Cheng et al., 2019). The observation localization gradually reduces the effect of an observation as the increasing departure from the analysis grid. In this study, the horizontal localization factor is calculated as the Gaussian function (Miyoshi et al., 2007):

$$f(r) = \exp(-r^2/2\sigma^2) \tag{8}$$

where $\sigma$ is the localization length and $r$ is defined as the physical distance between the observation and the analysis grid, and we force the localization factor to zero at 3.65 times the localization length (Zhao et al., 2015). In other word, we ignore observations beyond the cutoff distance. The tuneable horizontal and temporal localization lengths are defined in the physical distance (km) and time (hour), respectively. The vertical localization is not applied for the SO₂ emission inversion in this study, in other word, we trust the vertical profiles of the emission factors from the Model Inter-Comparison Study for Asia (MICS-Asia) phase III (Chen et al., 2019b).

A correct choice of the assimilation system parameters such as the ensemble size and correlation length is important to improve the data assimilation performance (Miyazaki et al., 2012b). A series of sensitivity experiments are performed to tune the assimilation system as listed in Table 1. A control experiment assuming the same emissions in November 2016 as in November 2010 (i.e., the standard MIX emissions) is conducted as the deterministic simulation to assess the influence of data assimilation. Considering the GOCART aerosol scheme uses a simple representation of the aerosol chemistry for reducing the computational load, we also conduct another deterministic simulation using a more sophisticated aerosol chemical scheme named Model for Simulating Aerosol Interactions and Chemistry (MOSAIC) coupled with the "lumped-structure" Carbon Bond Mechanism (CBMZ) (Zaveri et al., 2008) (i.e., chem_opt = 9) to investigate the effects of different chemistry and aerosol schemes on SO₂ oxidation. The data assimilation experiments are divided into three groups. In the first group, same random perturbation factor throughout the whole domain emission grids including vertical and temporal spaces per member is applied to the MIX SO₂ emission to generate 10 ensemble members for the WRF-Chem ensemble forward simulation. The spatial correlation coefficients among the initial prior ensemble of SO₂ emissions over every two model grids are equal to one, and this makes the spatial correlations among the grids points of the forecast emissions are also equal

to one. The same random perturbation factor generates a perfect correlation of emission in both the spatial and temporal spaces, however, this should not be seen as overly restrictive (Schutgens et al., 2010). Firstly, the "bottom-up" $SO_2$ emission inventories are to a large extent based on the used activity rates and emission factors (Li et al., 2018). Therefore, with the same random perturbation factors we effectively create an ensemble of inventories derived with different activity rates and emission factors. Secondly, the same emission standards for $SO_2$ emission mitigation are implemented in China (Zheng et al., 2018), and this induces the $SO_2$ reductions should be correlated at a certain extent in both spatial and temporal spaces. Thirdly, the analysis is conducted locally in 4D-LETKF, and the analysis at two grids separated by a distance over about 7.3 times the localization length is mostly independent (Schutgens et al., 2010). In this group, the strongest temporal localization is applied to assimilate only the observations within 1 hour of the local patch center. In other word, the hourly $SO_2$ emission is optimized using only the CNEMC $SO_2$ observation within the subsequent one hour, making the inverted $SO_2$ emission variable hour by hour. The difference of the experiments in this group is only the horizontal localization length, which is assumed as 10km, 30km, 50km, and 100km respectively. The purpose of the experiments in this group is to investigate the effects of horizontal localization length on $SO_2$ emission inversion. Based on the results in the first group as described latter in Section 4, the second group of experiments by fixing the horizontal localization length of 50km are subsequently performed with 10, 20, 40 ensemble members to investigate the effects of ensemble size on $SO_2$ assimilation. In this group, we remove the temporal localization to investigate the effects of temporal localization on the $SO_2$ emission inversion. In other word, the hourly $SO_2$ emission is optimized using all the CNEMC $SO_2$ observations within the 12 hours assimilation window, making the inverted $SO_2$ emission constant within every 12 hours. In the third group, the experiments are performed as same as those of the second group except that the ensembles are generated by independently perturbing the emission in horizontal space but dependently in vertical and temporal spaces. Those last two groups of experiments are used to investigate the effects of the ensemble size and perturbation factor on $SO_2$ emission inversion. The sensitivity data assimilation experiments are all performed for 10 days over the period of 00:00 UTC 8 November to 00:00 UTC 18 November 2016. The global model MOZART-4/GEOS-5 provides the initial and lateral boundary conditions used in this study (https://www.acom.ucar.edu/wrf-chem/mozart.shtml, last access: 10 August 2020). Since we don't know the uncertainties of the global model MOZART-4/GEOS-5, the initial and lateral boundary chemical fields are not perturbed in this study. The first three days are used as the spin-up of the data assimilation system, and the subsequent simulation results for one week are analysed in the next section. Based on the sensitivity tests of the $SO_2$ emission inversion system, the experiment H50kmT1hE10Ps, which generally performing better than other experiments, is extended to 00:00 UTC on 1 December 2016. This provides a longer period of 20 days to further validate the assimilation system. We also perform a recalculation experiment with the sophisticated CBMZ/MOSAIC scheme and the updated $SO_2$ emissions to verify the new $SO_2$ emission and the associated effects of $SO_2$ emission reduction.

**4 Results**

**4.1 Sensitivity of the inverted $SO_2$ emission to the assimilation parameters**

The spatial distribution of the MIX SO₂ emission in November 2010 at the model lowest layer is shown in Figure 3a, which is serving as the base of the initial priori SO₂ emission for our experiments in November 2016. The hotspots of the anthropogenic SO₂ emission are apparently found over the economically developed areas such as the North China Plain (NCP), the Yangtze River Delta (YRD), and the Peral River Delta (PRD). The Multi-resolution Emission Inventory for China (MEIC, http://www.meicmodel.org, last access: 15 February 2021) developed by Tsinghua University can provide the updated SO₂ emission in November 2016 (Fig. 3b), which is used as the independent "bottom-up" SO₂ emission to validate our inverted SO₂ emission. It is apparent that significant negative changes of SO₂ emission are found over the priori higher source regions such as the NCP, YRD, and PRD between the year of 2010 and 2016, which are in agreement with the changes of the column SO₂ concentrations observed by satellite (Wang et al., 2018). As the consequence of clean air actions (Zheng et al., 2018), the SO₂ emissions over most areas of China show systematic decline from the year 2010 to 2016 (Figure 4a). Can we reveal the reductions of the SO₂ emission by assimilating the CNEMC observed surface SO₂ concentration?

As shown in Figs. 3c-f and Figs. 4b-e, both spatial distribution and magnitude of the inverted SO₂ emission in November 2016 firstly become closer to the independent MEIC ones but get worse subsequently as increasing the horizontal localization length of the assimilation system. The inverted SO₂ emissions of each assimilation experiment are obtained by averaging the ones over the ensemble members. The spatial distributions of the mean differences of the MIX and inverted SO₂ emissions minus the MEIC ones are shown in Fig. S3 in the Supplement, and the spatial distributions of the mean ratios between the inverted SO₂ emissions and the MIX ones are shown in Fig. S4 in the Supplement. The time series of the hourly SO₂ emissions averaged over China of the initial MIX prior, the forecast and the analysis of the assimilation experiment H50kmT1hE10Ps from 00:00 UTC 8 November to 23:00 UTC 17 November 2016 are also shown in Fig. S5 in the Supplement, which illustrates the adjustment of SO₂ emissions with data assimilation. The experiment with the smallest horizontal localization length (i.e., 10 km) only optimizes the SO₂ emission over the specific grids where there are observations to be assimilated. In such a case, the significant reductions of the SO₂ emission over the grids with no observation sites are unable to be revealed, such as Shandong province in the NCP. With a larger localization length, an observation can constrain the emissions in more grids surrounding the observation and the observation error more gently increases as the distance from the observation location increased (Hunt et al., 2007). It is obvious that the systemic SO₂ emission reductions especially over the Shandong province are detected by enlarging the horizontal localization length. However, the perfect correlations of the emission perturbations over the domains with too large horizontal localization length cause spurious error covariance, inducing the more local emission changes undetectable. This is demonstrated as the inverted SO₂ emissions with a localization length of 100 km tend to lower than the independent MEIC ones with a mean bias of -0.44 $mol\ km^{-2}hr^{-1}$. Generally speaking, the inverted SO₂ emissions with horizontal localization length of 50 km are best in agreement with the MEIC ones with a mean bias of -0.15 $mol\ km^{-2}h^{-1}$ and Root Mean Square Error (RMSE) of 5.34 $mol\ km^{-2}hr^{-1}$.

With horizontal localization length of 50 km, the spatial distribution of the inverted $SO_2$ emission by removing the temporal localization is shown in Fig. 3g. It is clearly found that the inverted $SO_2$ emissions over the Shandong Province, YRD and PRD without temporal localization are lower than those with temporal localization, inducing larger negative bias and RMSE (Fig. 4f). It demonstrates that it is important to reveal the diurnal variations of the $SO_2$ emission (Wang et al., 2010). The experiment with temporal localization can reveal the hourly variation of the $SO_2$ emission by assimilating only the subsequent hourly observations, whereas the experiment without temporal localization only adjust the magnitude of $SO_2$ emission every 12 hours by assimilating all the observations within the 12-hour window.

As shown in Figs. 3g-i and Figs. 4f-h, there are no significant differences of the horizontal distribution and magnitude of the inverted $SO_2$ emission between 10, 20 and 40 ensemble members. This indicates that the ensemble size has little effect on the $SO_2$ emission inversion when randomly correlated perturbing the emissions. The ensemble forecast with 10 members seems feasible to reveal the $SO_2$ reductions in China, although the inverted emissions have not converged properly. This in turn significantly reduces the required computational resources and time for the forward calculation of the ensemble model, making the dynamical update of air pollutant emissions affordable when assimilating near-real-time observations.

The inverted $SO_2$ emission with horizontal random uncorrelated perturbations gets closer to the independent MEIC one as increasing the size of the ensemble member (Figs. 3j-l and Figs. 4i-k). However, the performances of the horizontal distribution and magnitude of the inverted $SO_2$ emission using 40 ensemble members with horizontal random uncorrelated perturbations are even obviously worse than those using 10 ensemble members with horizontal correlated perturbations. It demonstrates that the independent emission perturbations over each model grid tend to underestimate the model spread due to the current limited ensemble members and the cancellation of neighbouring cells (Pagowski and Grell, 2012;Schutgens et al., 2010).

The mean bias and RMSE of $SO_2$ emission over China by using the MIX $SO_2$ emission in November 2010 for that in November 2016 are 2.70 and 9.78 $mol\ km^{-2}h^{-1}$, respectively (Fig. 4a). For the inverted $SO_2$ emission by data assimilation, the bias and RMSE reduction rates (Miyazaki et al., 2012b) are estimated as follows,

$$\frac{2.70 - |B_{DA}|}{2.70} \times 100. \qquad (9)$$

$$\frac{9.78 - |RMSE_{DA}|}{9.78} \times 100. \qquad (10)$$

where $B_{DA}$ and $RMSE_{DA}$ are the mean bias and RMSE between the inverted $SO_2$ emission and the MEIC $SO_2$ emission in November 2016. As shown in Fig. 5, it is obviously found that (1) the inverted $SO_2$ emission in every assimilation experiment can both reduce the bias and RMSE; (2) the randomly correlated perturbation factor is superior to the randomly uncorrelated perturbation factor in reducing the bias and RMSE, and it is generally unaffected by the ensemble size; (3) the experiment H50kmT1hE10Ps shows the best performance in both reducing the bias and RMSE, decreasing the bias and RMSE by 94.5% and 45.4% respectively.

## 4.2 Sensitivity of the surface SO₂ concentration to the emission

Figures 6 and 7 show the horizontal distributions of the biases and RMSEs between the surface $SO_2$ concentrations simulated in various experiments and the CNEMC observed ones over both the assimilated and independent sites. The $SO_2$ concentrations in each assimilation experiment are obtained by averaging the ones over the WRF-Chem ensemble recalculations with the optimized emissions. The spatial distributions of the mean $SO_2$ concentrations simulated with the original MIX emissions and the updates of the simulated $SO_2$ concentrations with the inverted $SO_2$ emissions are shown in Fig. S6 in the Supplement. The spatial distribution of the mean differences of the $SO_2$ concentrations simulated in the FR and FR_CM experiments are also shown in Fig. S6 in the Supplement. It is apparent that significant RMSEs and positive biases are found over the priori $SO_2$ emission hotspot regions such as the NCP, YRD, and PRD in both the two free run experiments, whereas slight RMSEs and negative biases are both found over northwestern China. Furthermore, the horizontal distributions of both the biases and RMSEs of the two free run experiments are generally similar. As given in Table 2, the relative differences of the RMSEs in the FR and FR_CM experiments are both less than 1% over the assimilated and independent sites, although the mean biases in the FR_CM experiment tend to both slightly smaller than those in the FR experiment. Those demonstrate that the biases and RMSEs between the simulated and observed surface $SO_2$ concentrations are not induced by the uncertainties of the different chemical reaction mechanisms but due to the uncertainties of the used $SO_2$ emissions. The simulated ensemble mean surface $SO_2$ concentrations by recalculating the WRF-Chem with the inverted $SO_2$ emissions in all assimilation experiments show more comparable to the observations, and the performances of the simulated $SO_2$ surface concentrations are clearly affected by the inputs of the different inverted $SO_2$ emissions due to assimilation system parameters. This indicates that the uncertainties of the different chemical reaction mechanisms in simulating $SO_2$ concentrations are much smaller than those of the $SO_2$ emissions. In the first group of data assimilation experiments, the largest biases and RMSEs of the simulated and observed $SO_2$ surface concentrations over both the assimilated and independent sites are found in the H10kmT1hE10Ps experiment. This indicates that the $SO_2$ emission changes exist grid correlations and the $SO_2$ emission inversions over only the grids with available assimilated sites are not sufficient to reveal the real $SO_2$ emission changes in the grids without observation sites. In addition, largest biases and RMSEs over both the assimilated and independent sites are still found in the third group of data assimilation experiments, although the biases and RMSEs are decreasing as the increasement of the ensemble members. This further illustrates there are correlations of the grided $SO_2$ emission changes and the random perfect correlated emission perturbation factors over the model grids are superior to the random uncorrelated emission perturbations for current emission inversions. The latter is probably due to the current limited ensemble members for reducing the computational resources. However, the sophisticatedly random uncorrelated emission perturbations should have better performances with large or unlimited ensemble members. Similar to the inverted emissions, the experiments in the second group show the ensemble size has little effects on the biases and RMSEs of the $SO_2$ surface concentrations over both the assimilated and independent sites when the ensemble members are generated by perturbing the emissions perfect correlated over the domain grids. The reductions of the biases of the $SO_2$ surface concentrations in both the assimilated and independent sites are benefitted from the temporal

localization, although the RMSEs are slightly increased. It is interesting that the smallest RMSE of the $SO_2$ surface concentrations over the independent sites is also found in the H50kmT1hE10Ps experiment with value of 36.20, which the inverted $SO_2$ emissions are also best in agreement to the independent MEIC ones. This further indicates the assimilation system parameters used in this experiment are suitable for the $SO_2$ emission inversion, decreasing the biases of $SO_2$ surface concentrations over assimilated and independent sites by 87.2% and 88.9% respectively. The underestimation of the surface $SO_2$ concentration with the original MIX emission over northwestern China such as the Gansu province is potentially attributable to the increasing $SO_2$ emissions due to energy industry expansion and relocation over northwestern China (Ling et al., 2017). The $SO_2$ emissions and surface concentrations over the Gansu province are increased to reduce the negative biases in the assimilation experiments as shown in Figs. S4 and S6 in the Supplement, indicating our emission inversion system also works well when the prior emissions are underestimated. However, the simulated surface $SO_2$ concentrations with the inverted emissions are still underestimated over the Gansu province. The reason for the underestimation is twofold: (1) there are limited observations to be assimilated over northwestern China because the observation sites are sparse; (2) the initial priori MIX $SO_2$ emission over northwestern China is small and underestimated, inducing the model uncertainty is small relative to the observation one. This translates to a reduced impact of the observation on the priori emission.

Figure 8 illustrates the frequency distributions of the deviations of the simulated $SO_2$ surface concentrations in various experiments minus the observed ones. It is expected that the distributions of the $SO_2$ surface concentrations deviations for the two free run experiments in China and the three subregions are all positively biased due to the known overestimation of the $SO_2$ emissions. The distributions of the $SO_2$ surface concentration deviations with the updated $SO_2$ emissions in all the data assimilation experiments show reduced biases over both the assimilated and independent sites. However, the distributions of the deviations with the updated $SO_2$ emissions in the third group of experiments and the H10kmT1hE10Ps experiment are still positively biased, whereas slightly negative biased are found in the second group of experiments and the H100kmT1hE10Ps experiment. The distributions of the $SO_2$ concentration deviations with the updated $SO_2$ emissions in the H50kmT1hE10Ps experiment, as expected, shows the best performance with a peak closer to 0 in both the assimilated and independent sites.

### 4.3 $SO_2$ reduction in China and associated effects

Based on the aforementioned sensitivity tests of the $SO_2$ emission inversion system, the experiment H50kmT1hE10Ps is extended to 00:00 UTC on 1 December 2016. This provides a longer period for 20 days to estimate the reduction of the $SO_2$ emission over China in November over the period 2010-2016. The Bottom-up and Top-down estimations of the $SO_2$ emission reduction from 2010 to 2016 are calculated by comparing the MEIC and inverted $SO_2$ emissions by data assimilation in November 2016 to the MIX $SO_2$ emission in November 2010. As shown in Figure 9, the Top-down estimation of the $SO_2$ emission reduction over China is 49.4%, which is well agreement with the Bottom-up estimation of 48.0%. In addition, larger $SO_2$ emission reductions over the three subregions estimated by the Bottom-up approach are correctly revealed by the emission inversion system. The Top-down and Bottom-up estimations of the $SO_2$ emission

reduction over NCP are generally comparable with values of 56.0% and 52.4% respectively. The largest $SO_2$ emission reductions both with the Top-down and Bottom-up approaches are found over the YRD region with values of 73.1% and 61.8% respectively. The $SO_2$ emission reduction by Top-down approach are 10% higher than that by the Bottom-up approach over the PRD region. To validate the inverted $SO_2$ emissions and explore the possible reasons of the overestimation

of $SO_2$ emission reduction over the YRD and PRD by Top-down approach, the time series of the simulated $SO_2$ surface concentrations in various experiments and the observed ones are shown in Figure 10. The simulated $SO_2$ surface concentrations especially over the YRD subregion in the two free run experiments show significant positive biases over all the period, revealing the drawback of the prescribed $SO_2$ emissions in November 2016 as same as that in November 2010. The simulated $SO_2$ surface concentrations with the inverted $SO_2$ emissions using both the RADM2/GOCART and

CBMZ/MOSAIC chemical reaction mechanisms are much closer to the observations in both the assimilated and independent sites over all the period. It demonstrates the WRF-Chem/4D-LETKF emission inversion system can continuously and dynamically update the $SO_2$ emissions by assimilating the newly available observations as shown Fig. S5 in the Supplement. The $SO_2$ surface concentrations simulated by the FR_CM experiment are sometime lower than those in the FR experiment especially over the YRD and PRD subregions, indicating the overestimations of the $SO_2$ emission reduction by the Top-

down approach over the YRD and PRD are probability due to the simple aerosol chemistry schemes used in RADM2/GOCART (Chin et al., 2000). This is proved as the simulated $SO_2$ surface concentrations in YRD with the RADM2/GOCART scheme and the inverted $SO_2$ emissions over the period 18-22 November 2016 are generally comparable to the observed ones, whereas the simulated $SO_2$ surface concentrations with the sophisticated CBMZ/MOSAIC scheme and the inverted $SO_2$ emissions are lower than the observed ones. The simulated $SO_2$ surface concentrations at all sites with the

inverted emission in both the FR_CM and assimilation recalculation are generally underestimated. This is due to the inverted emission is sufficient to reduce the overestimations of $SO_2$ concentration over the priori $SO_2$ emission hotspot regions but insufficient to eliminate the underestimations over northwestern China.

Based on the inverted $SO_2$ emissions from 11 November to 1 December 2016, the daily and diurnal variations of the $SO_2$ emission reductions over China and the NCP subregion are estimated as shown in Figs. 11a and b respectively, and the

diurnal variations of the inverted $SO_2$ emissions over China and the NCP subregion are also shown in Fig. 11c. Generally speaking, the daily variation of the $SO_2$ emission reduction over China is not so significant. Larger $SO_2$ emission reductions over the period 17 to 19 November in the NCP induced by the first orange alert for heavy winter air pollution in 2016 are clearly detected by the inverted emissions (Shi et al., 2019). Lower $SO_2$ emission reductions over China and NCP from 21 to 22 November are probably contaminated by the strong cold wind from the northwestern direction, inducing the lowest $SO_2$

concentrations and underestimating the associated ensemble spread. The latter induces the inverted emission to be overconfident in the background emission (Hunt et al., 2007). Since the emissions are constant over time in the priori MIX inventory, the diurnal variations of the $SO_2$ emission reduction over China and NCP both reveal higher emission reductions in the nighttime, inducing the $SO_2$ emissions in the nighttime are lower than those in the daytime (Fig. 11c). This is generally reasonable as less human and economic activities  happen in the nighttime (Chen et al., 2019b).

Figure 12 shows the spatial distributions of the averaged surface concentrations of the sulfate, ammonium, nitrate, and PM$_{2.5}$ over 11 November to 1 December 2016 simulated with the CBMZ/MOSAIC mechanism and the original MIX emissions, and the absolute and relative changes of the associated aerosol surface concentrations with the newly inverted emissions by data assimilation. It is found that the SO$_2$ emission reductions induce the sulfate surface concentrations reduced up to 10 $\mu g/m^3$ (50%) over the center China, and this is due to the sulfate aerosols are dominated by the productions in-cloud

oxidations (Chin et al., 2000;Goto et al., 2015) and more cloud are found over the center China (Li et al., 2015;Ma et al., 2014). The nitrate surface concentrations are found slightly increased in the center China as the reductions of sulfate aerosols, and this is due to the emissions of the nitrate precursors (i.e., NO and NO$_2$) are not updated in this study and NH$_4$NO$_3$ is formed only in sulfate-poor aerosols (Zaveri et al., 2008;Chen et al., 2016). The synergy effects of sulfate-nitrate-ammonium induce slightly reductions of ammonium surface concentrations, decreasing the PM$_{2.5}$ surface concentrations

about 10 $\mu g\ m^{-3}$ (10%) over the center China.

**5 Conclusions**

The timely precise emission inventories are crucial to air quality prediction and mitigation. To dynamically update the emissions of air pollutants, we have developed a new emission inversion system based on the 4D-LETKF and the fully

coupled model named WRF-Chem. Our emission inversion system considers the complex nonlinear relationship between atmospheric chemical concentrations and emissions by ensemble forecasting with perturbed emissions. The emission inversion system is examined to update the outdated MIX SO$_2$ emissions in November 2010 by assimilating the quality-assured and controlled observations of SO$_2$ surface concentration from the CNEMC in November 2016. The inverted SO$_2$ emissions over China by data assimilation for November 2016 are validated with the independent MEIC emissions in

November 2016.

Sensitivity tests for the emission inversion system demonstrate that the assumption of the covariance error matrix of the a priori SO$_2$ emissions has the largest effect on the inverted emissions. The random perfectly correlated emission perturbations throughout the whole model grids with horizontal localization length of 50 km can best reproduce the independent MEIC SO$_2$ emissions, decreasing the MIX emission bias and RMSE by 94.5% and 45.4% respectively. The independent emission

perturbations over each model grid tend to underestimate the model spread due to the current limited ensemble members and the cancellation of neighbouring cells. With the random perfectly correlated emission perturbations, the ensemble size has only little effect on the inverted SO$_2$ emissions and the ensemble forecast with 10 members seems feasible to reveal the SO$_2$ reductions in China. The temporal localization by assimilating only the subsequent hourly observations can reveal the diurnal variation of the SO$_2$ emission, which is better than that to update the magnitude of SO$_2$ emission every 12 hours by

assimilating all the observations within the 12-hour window.

The known overestimates of the prescribed SO$_2$ emissions in November 2016 as same as that in November 2010 are successfully detected as the simulated SO$_2$ surface concentrations especially over the SO$_2$ emission hotspot subregions with two distinguished chemical reaction mechanisms are both significantly positive biased. The simulated SO$_2$ surface

concentrations with the inverted $SO_2$ emissions in all assimilation experiments show more comparable to the observations, and the performances of the simulated $SO_2$ surface concentrations are clearly affected by the inputs of the different inverted $SO_2$ emissions due to assimilation system parameters. This indicates that the uncertainties of the different chemical reaction mechanisms in simulating $SO_2$ concentrations are much smaller than those of the $SO_2$ emissions. The smallest RMSE of the simulated and observed $SO_2$ surface concentrations over the independent verification sites is also found in the experiment that the inverted $SO_2$ emissions are best in agreement to the independent MEIC ones, decreasing the biases of $SO_2$ surface concentrations by 88.9%.

The $SO_2$ emission reduction over China in November over the period 2010 to 2016 is estimated as 49.4% by assimilating the observations of surface $SO_2$ concentrations, which is well agreement with the Bottom-up estimation of 48.0%. In addition, larger $SO_2$ emission reductions over the NCP, YRD and PRD estimated by the Bottom-up approach are correctly revealed by the emission inversion system. Largest $SO_2$ emission reductions both with the Top-down and Bottom-up approaches are found over the YRD region with values of 73.1% and 61.8% respectively, and the simple parameterizations of the aerosol chemistry in the GOCART scheme may induce the overestimates of the $SO_2$ emission reductions about 10 percent. The $SO_2$ emission reductions induce the sulfate and $PM_{2.5}$ surface concentrations to decrease up 10 $\mu g\ m^{-3}$ over the center China.

**Code/Data availability.** The WRF-Chem and LETKF source codes are available with the terms and conditions in https://www2.mmm.ucar.edu/wrf/users/download/get_source.html and https://github.com/takemasa-miyoshi/letkf. The MIX Asian emission inventory can be found in http://meicmodel.org/dataset-mix.html.

**Author contributions.** TD designed and performed the experiments used in the study. TD and YC conducted the data analysis. TD prepared the manuscript with help from YC, DG, GS, and TN. DG and TN provided the super-computer resources.

**Competing interests.** The authors declare that they have no conflict of interest.

**Acknowledgments.** This research has been supported by the National Key Research and Development Program of China (grant nos. 2017YFC0209803 and 2016YFC0202001), the Strategic Priority Research Program of the Chinese Academy of Sciences (grant no. XDA2006010302), the National Natural Science Funds of China (grant nos. 41875133, 41605083, 41905119, and 41590875), and the Youth Innovation Promotion Association CAS (2020078). We are grateful to the relevant researchers who provided NCEP FNL reanalysis data (https://rda.ucar.edu/datasets/ds083.2/). The model simulations were performed using the supercomputer resources NIES/HPE Apollo 2000.

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

**Figures**

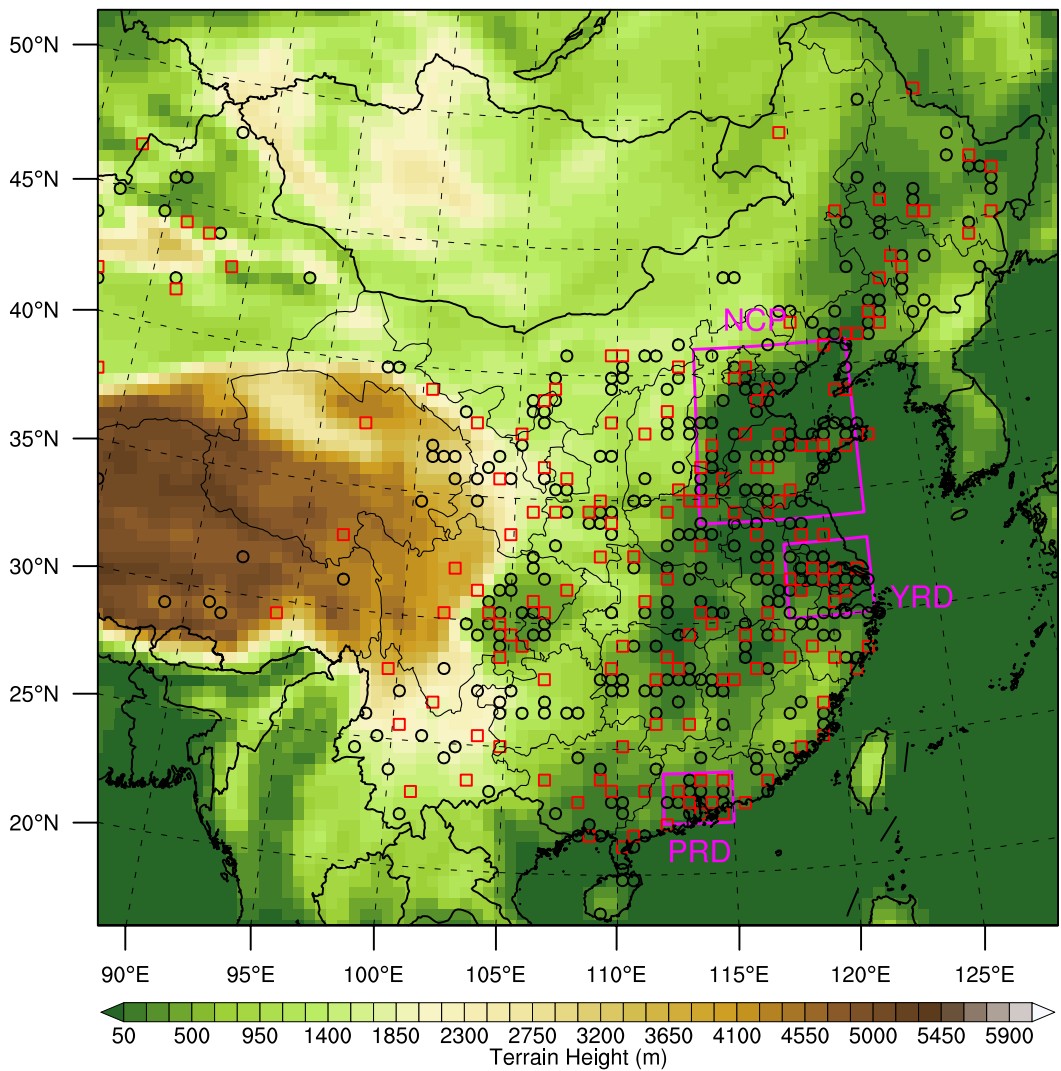

Figure 1. WRF-Chem model computational domain with the topography. The locations of the assimilated and independent verification observation sites of the China National Environmental Monitoring Centre (CNEMC) are shown as the black circles and red squares, respectively. The three magenta boxes mark the North China Plain (NCP), the Yangtze River delta (YRD) and the Peral River delta (PRD) subregions where relatively dense observation sites are available.

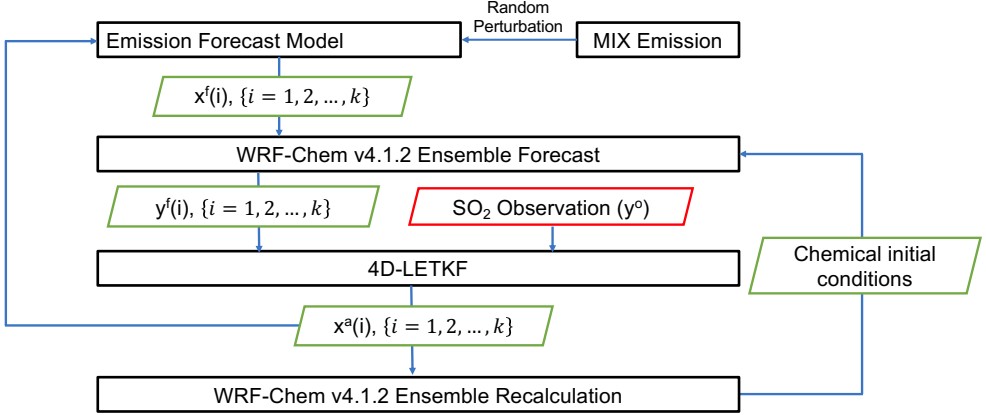

Figure 2. Flowchart of the WRF-Chem/4D-LETKF $SO_2$ emission inversion system by assimilating the $SO_2$ observations.

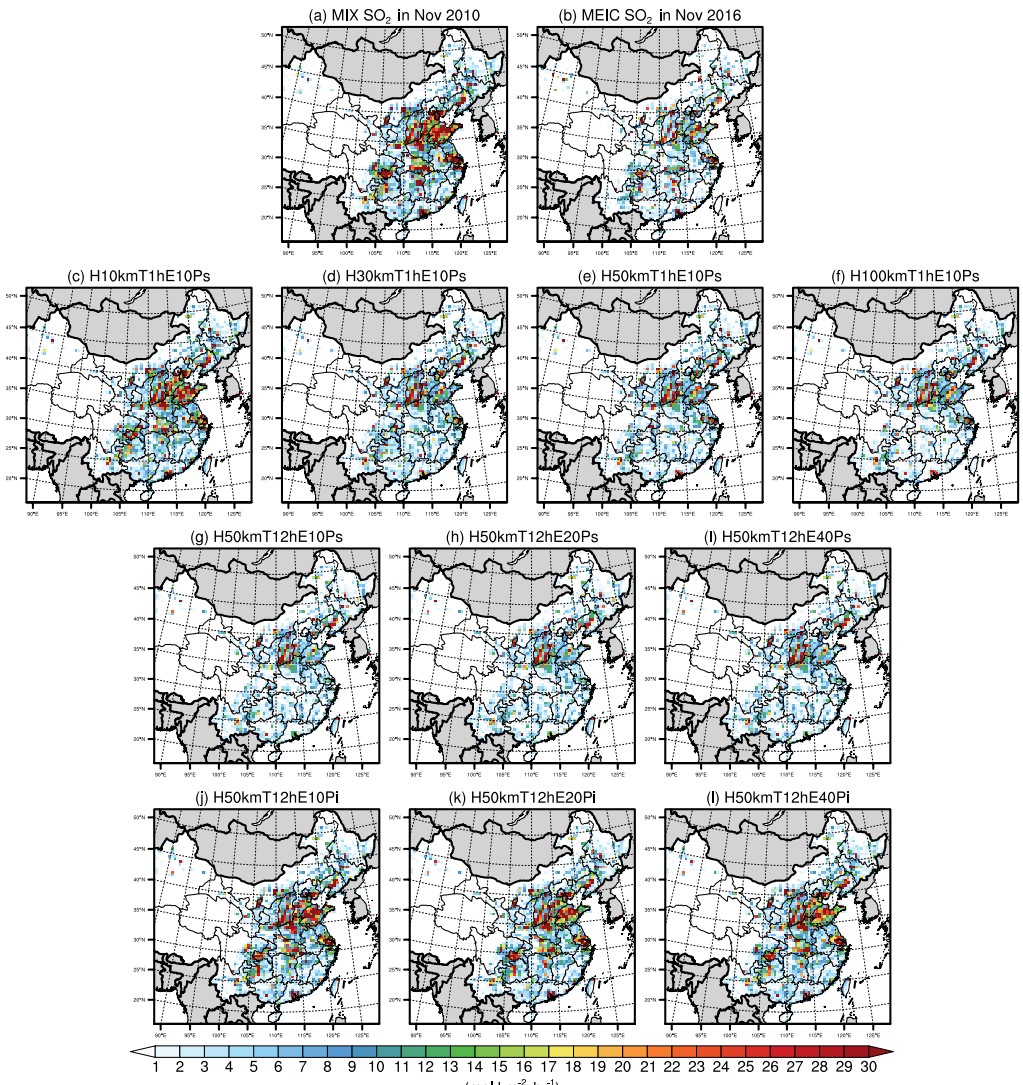

Figure 3. Spatial distributions of the MIX SO$_2$ emission in November 2010 (a) and the MEIC SO$_2$ emission in November 2016 (b) at the model lowest layer. Spatial distributions of the inverted SO$_2$ emissions in November 2016 in various data assimilation experiments (c-l).

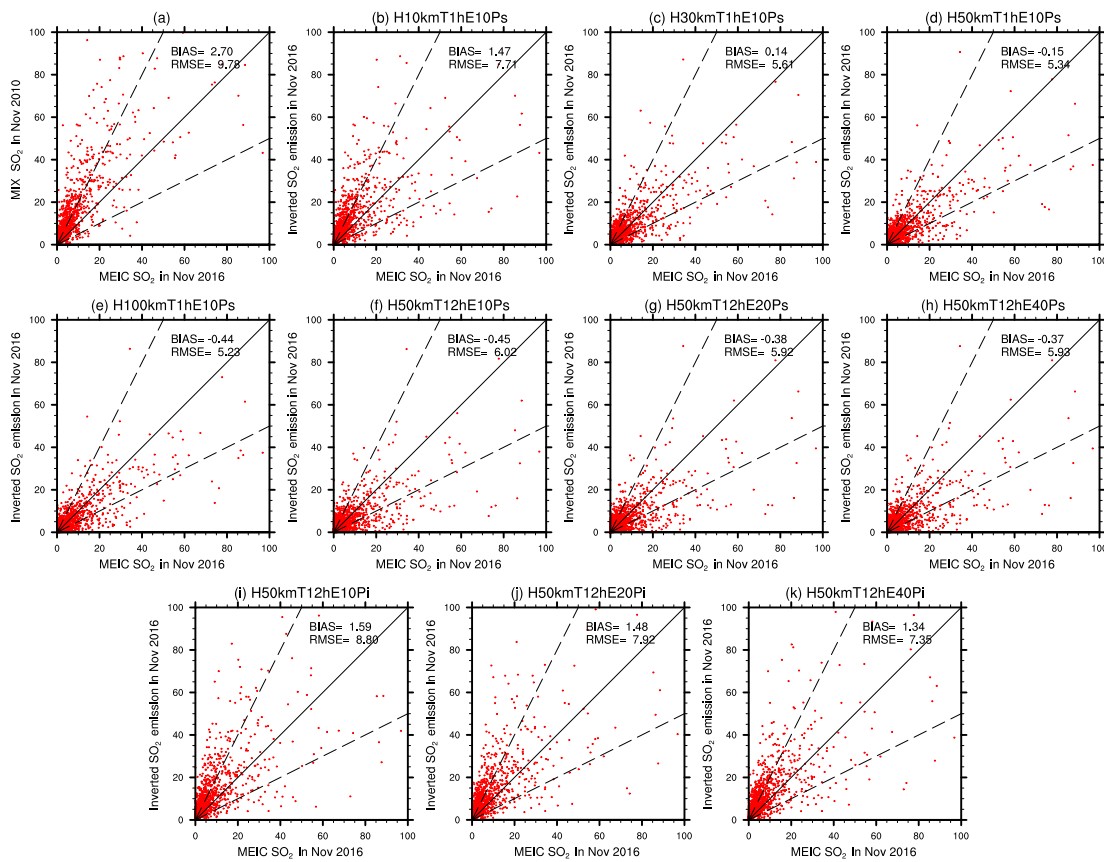

Figure 4. Comparisons of the MIX SO$_2$ emissions in November 2010 (a) and the inverted SO$_2$ emission in November 2016 in various data assimilation experiments (b-k) to the MEIC SO$_2$ emissions in November 2016.

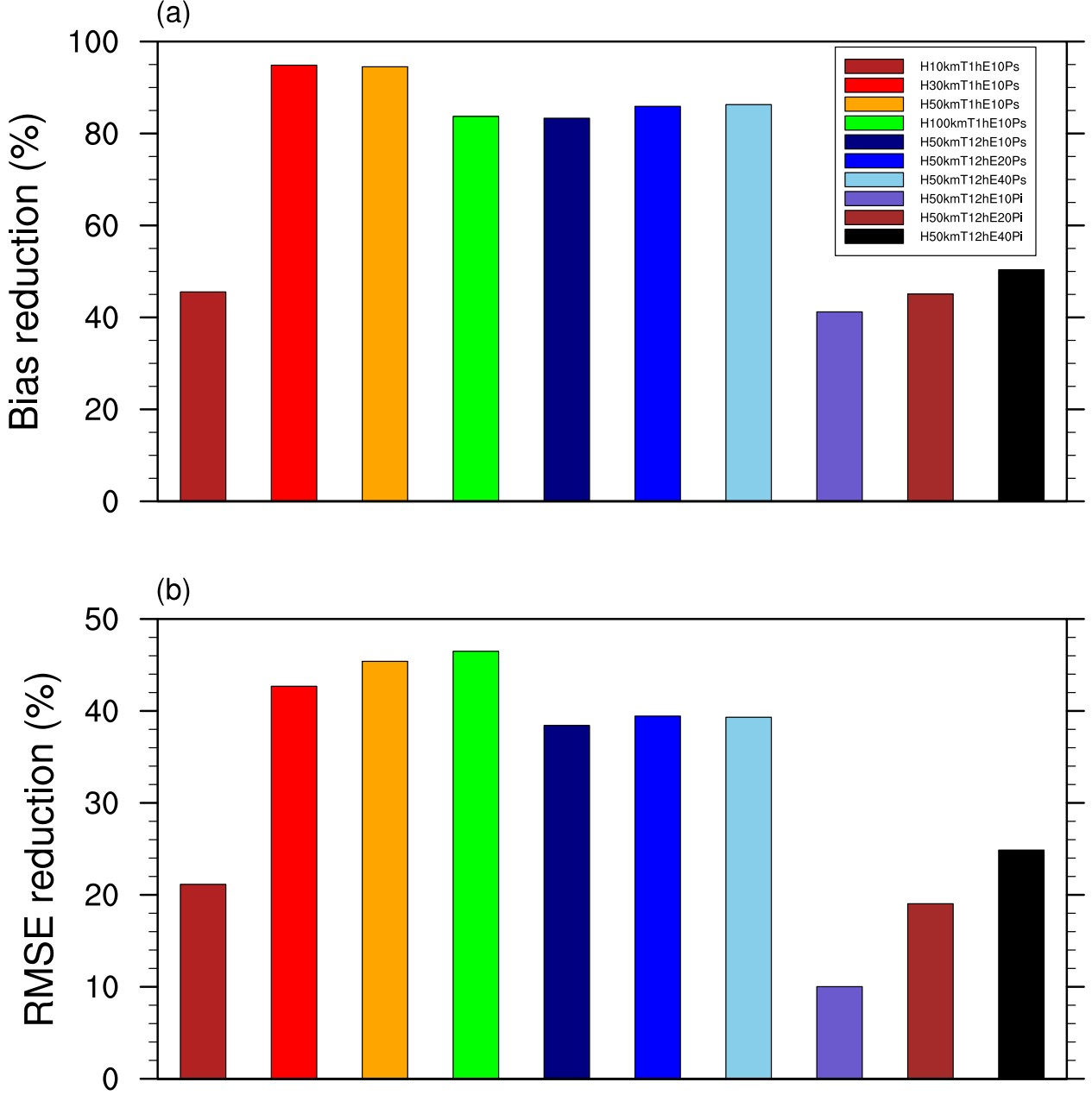

Figure 5. Reductions of the bias and Root Mean Square Error (RMSE) between the inverted SO$_2$ emissions in various data assimilation experiments and the MEIC ones referring to those between the MIX and MEIC SO$_2$ emissions.

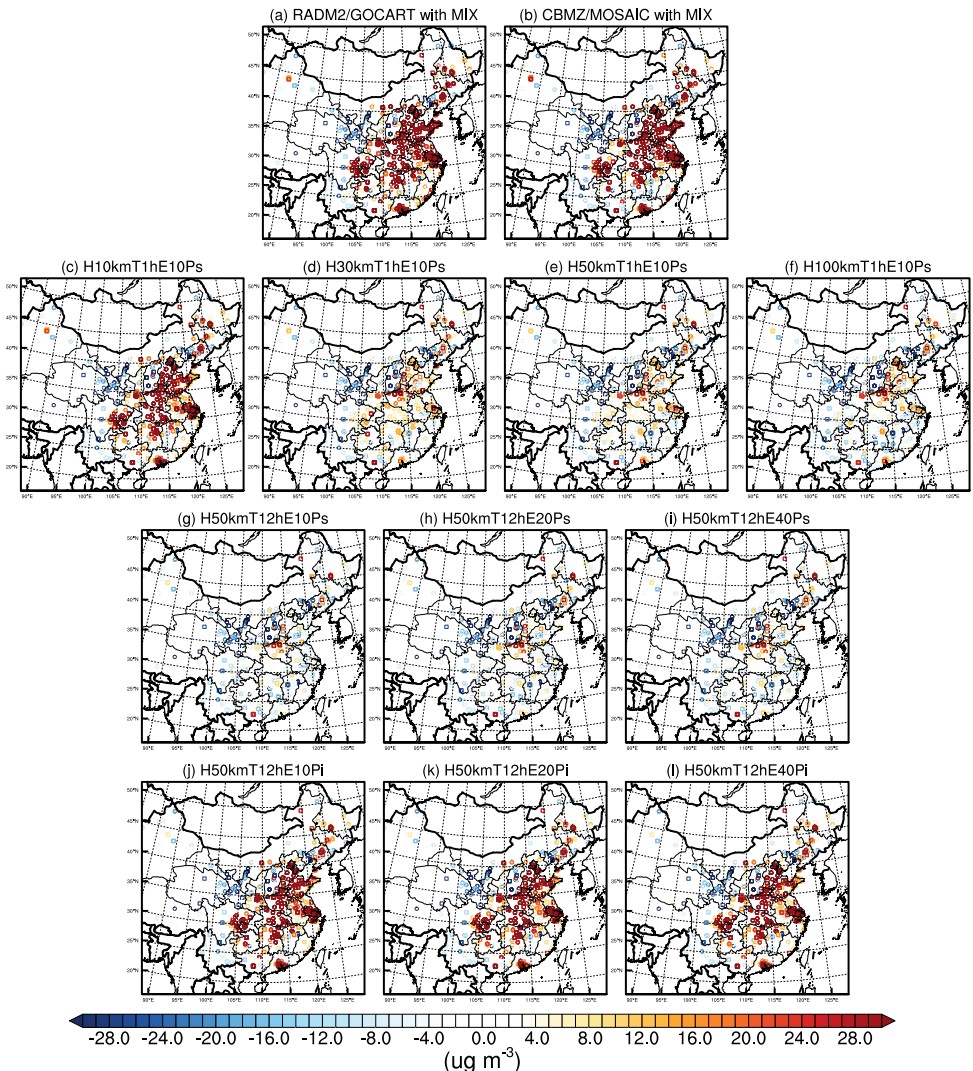

Figure 6. Spatial distributions of the mean biases between the simulated surface SO$_2$ concentrations in various experiments and the CNEMC observed ones over both the assimilated and independent sites. The locations of the assimilated and independent verification observation sites of the CNEMC are shown as the circles and squares, respectively.

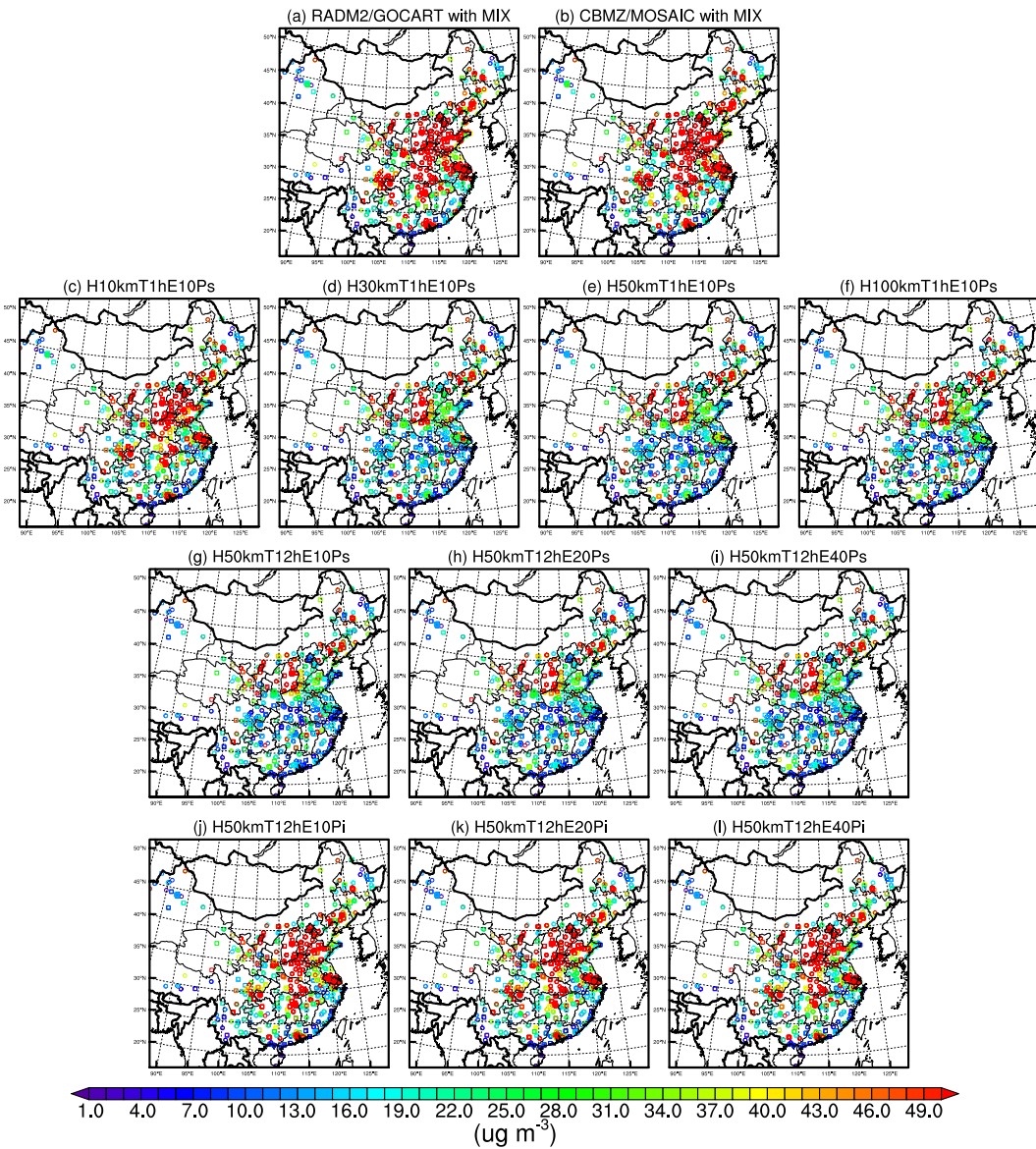

Figure 7. Same as Figure 6 but for the RMSEs.

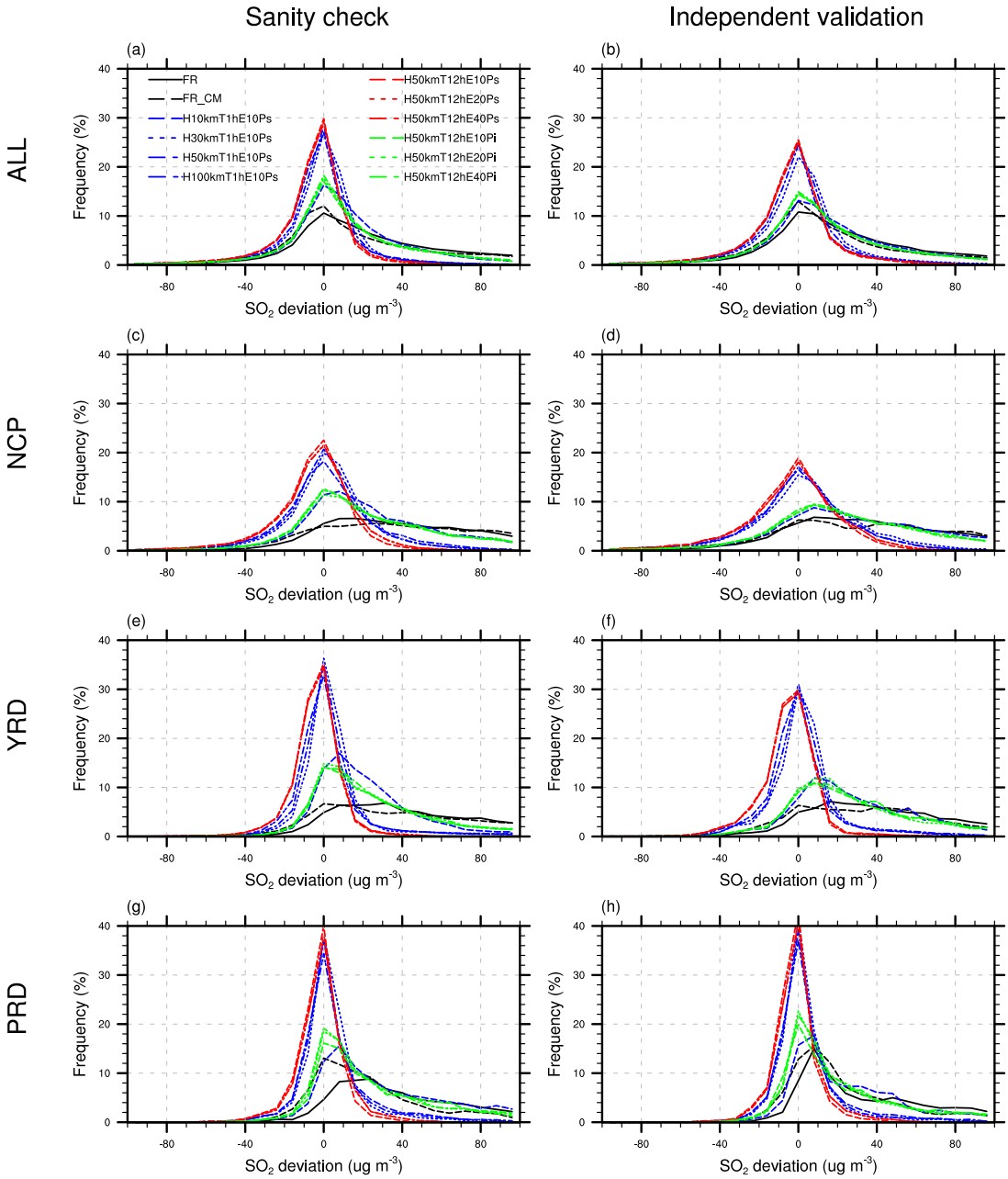


Figure 8. Frequency distributions of the deviations of the simulated $SO_2$ surface concentrations in various experiments minus the observed ones.

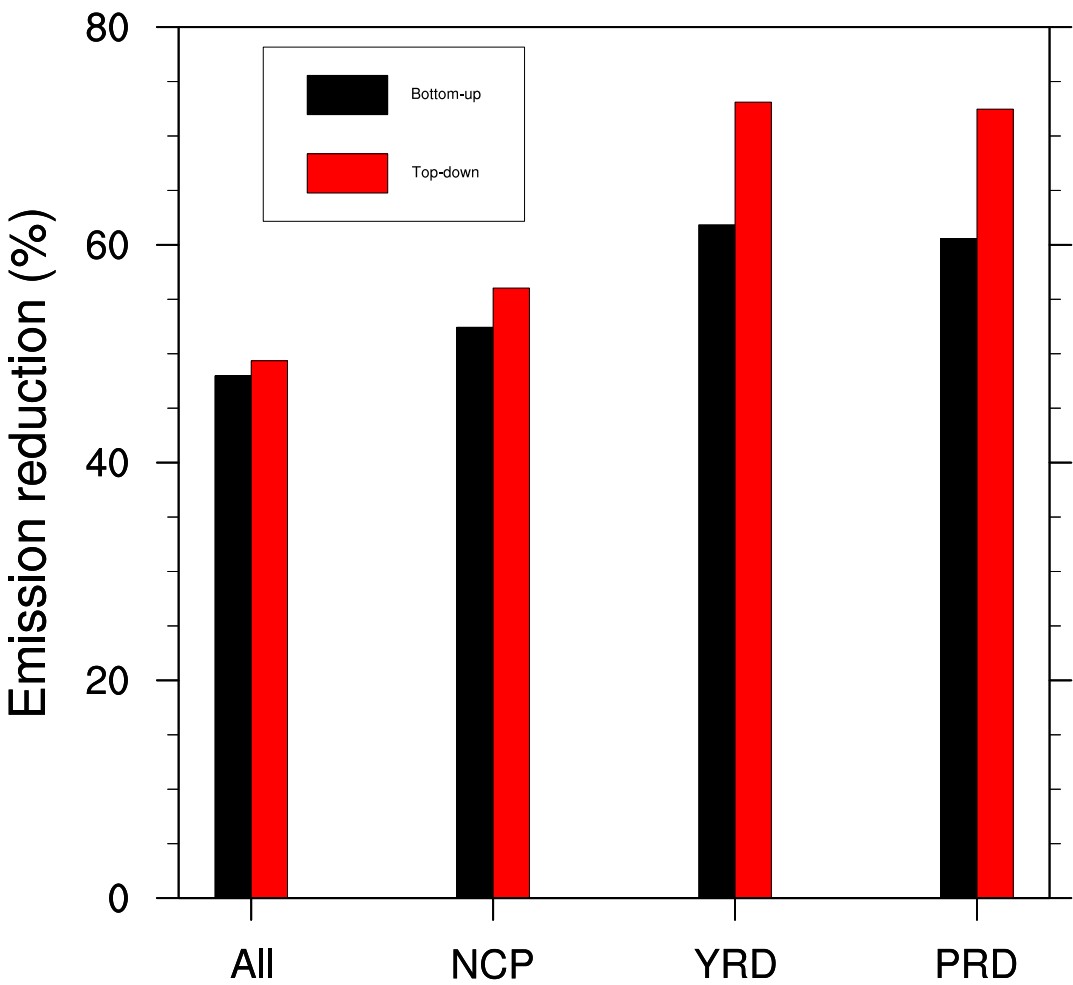

Figure 9. SO$_2$ emission reductions in November over the period 2010 to 2016 in China and the three subregions estimated by
the Bottom-up and Tow-down approaches.

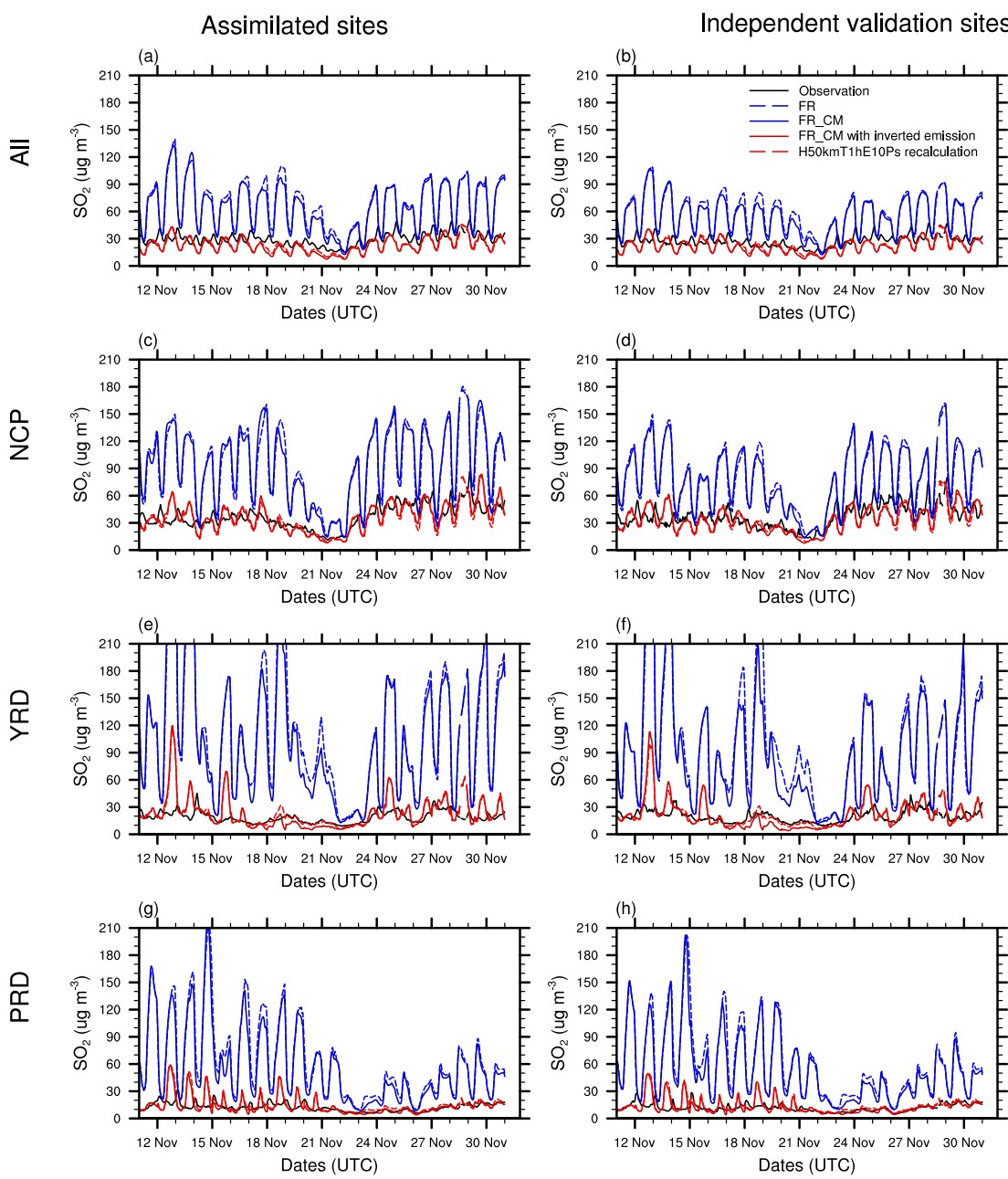

Figure 10. Time series of the simulated $SO_2$ surface concentrations in various experiments and the CNEMC observations.

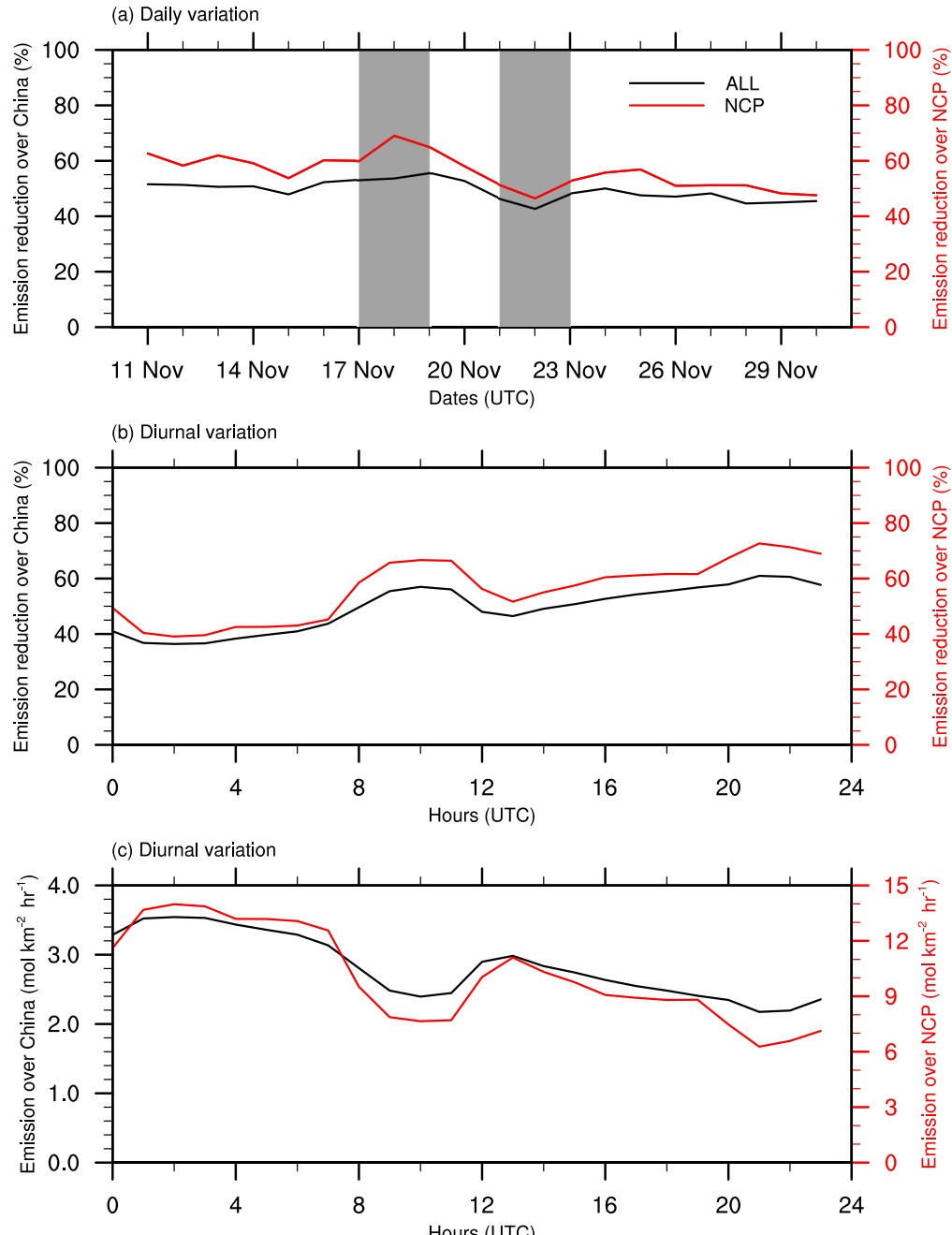

Figure 11. Daily (a) and diurnal (b) variations of the $SO_2$ emission reductions over China and the NCP subregion based on
the inverted emissions. Diurnal variations of the inverted $SO_2$ emissions over China and the NCP subregion (c).

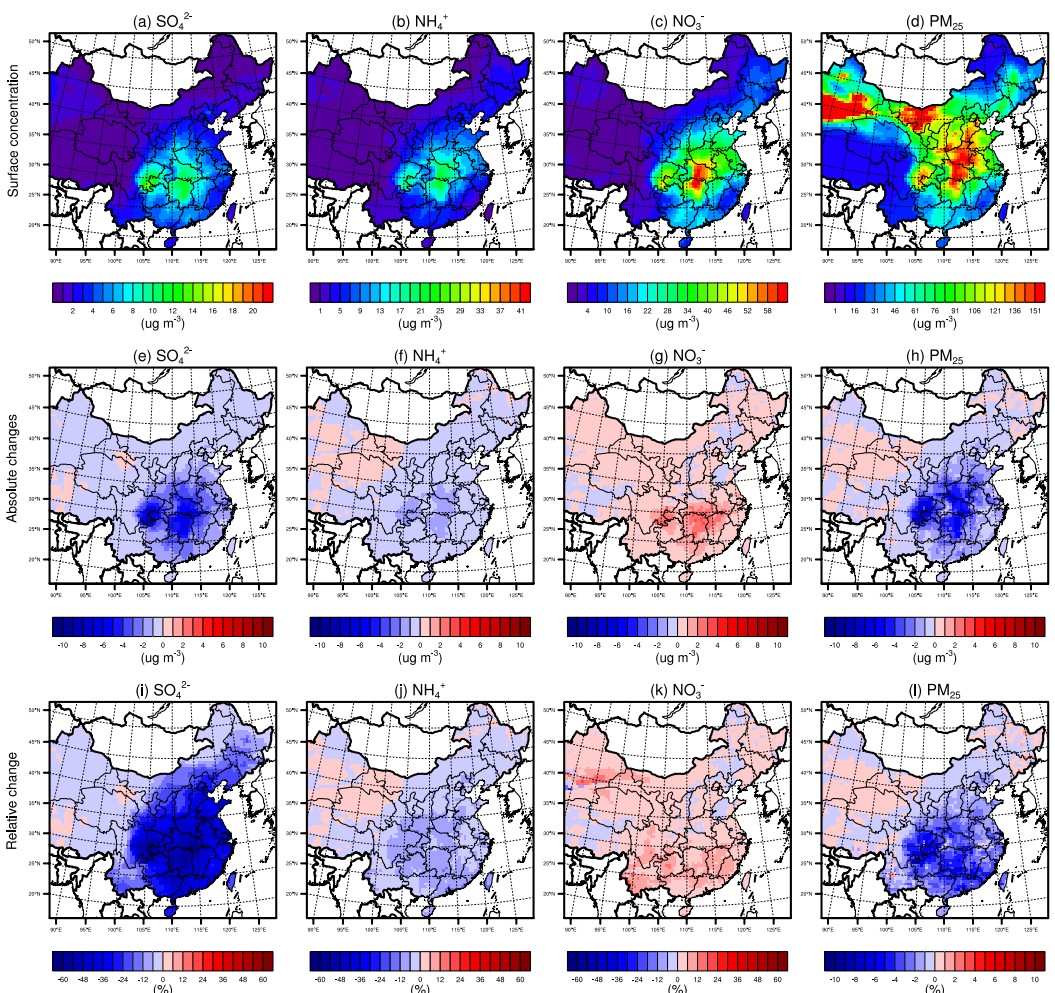

Figure 12. Spatial distributions of the averaged surface concentrations of the sulfate, ammonium, nitrate and PM2.5 over 11 November to 1 December 2016 simulated with the CBMZ/MOSAIC mechanism and the MIX emissions in November 2010, and the absolute and relative changes of the associated aerosol surface concentrations with the updated emissions by data assimilation.



## Tables

**Table 1.** Experimental design in this study.

| Experiments | | Design and purpose of the simulation |
|---|---|---|
| Control experiments | FR<br><br>FR_CM | Free run using RADM2/GOCART (FR) and CBMZ/MOSAIC (FR_CM) mechanisms with the MIX emission inventory in November 2010 to investigate the effects of different chemistry and aerosol schemes on $SO_2$ simulations and provide a reference to evaluate the effects of data assimilation |
| Data assimilation experiments | H10kmT1hE10Ps<br>H30kmT1hE10Ps<br>H50kmT1hE10Ps<br>H100kmT1hE10Ps | Same random perturbation factors throughout the whole domain emission grids and 12 hours to generate 10 ensembles with assimilated observations within 1h. Experiments with horizontal localization length of 10km, 30km, 50km, and 100km respectively are performed to investigate the effects of horizontal localization length on $SO_2$ emission inversion. |
| | H50kmT12hE10Ps<br>H50kmT12hE20Ps<br>H50kmT12hE40Ps | Same random perturbation factors throughout the whole domain emission grids and 12 hours to generate 10, 20, 40 ensembles with assimilated observations within 12 h and horizontal localization length of 50km based on above tests. Experiments in this group are performed to investigate the effects of ensemble size on $SO_2$ emission inversion. |
| | H50kmT12hE10Pi<br>H50kmT12hE20Pi<br>H50kmT12hE40Pi | Horizontal independent random perturbation factors in each emission grid but same throughout 12 hours to generate 10, 20, 40 ensembles with assimilated observations within 12 h and horizontal localization length of 50km. Experiments in this group together with above group are performed to investigate the effects of ensemble size and perturbation factors on $SO_2$ emission inversion. |
| Recalculation experiment | CBMZ/MOSAIC with posterior emission | Deterministic simulation with sophisticated CBMZ/MOSAIC scheme is recalculated with the updated $SO_2$ emission to verify the updated $SO_2$ emissions with an independent mechanism and the associated effects of $SO_2$ emission reduction. |


**Table 2.** The mean biases and Root Mean Square Errors (RMSEs) of the simulated $SO_2$ surface concentrations in various experiments and the CNEMC observed ones over all assimilated and independent sites.

| Experiments | Sanity Check | | Independent Validation | |
|---|---|---|---|---|
| | B | RMSE | B | RMSE |
| FR | 44.03 | 106.04 | 34.72 | 78.03 |
| FR_CM | 36.49 | 106.69 | 27.01 | 77.45 |
| H10kmT1hE10Ps | 12.01 | 50.89 | 21.83 | 59.84 |
| H30kmT1hE10Ps | -4.06 | 38.20 | -0.34 | 38.57 |
| H50kmT1hE10Ps | -5.65 | 38.63 | -3.84 | 36.18 |
| H100kmT1hE10Ps | -7.36 | 39.66 | -5.76 | 36.20 |
| H50kmT12hE10Ps | -9.75 | 37.36 | -7.45 | 34.59 |
| H50kmT12hE20Ps | -8.80 | 37.54 | -6.20 | 35.27 |
| H50kmT12hE40Ps | -8.75 | 37.55 | -6.15 | 35.28 |
| H50kmT12hE10Pi | 20.42 | 75.96 | 21.62 | 65.75 |
| H50kmT12hE20Pi | 17.38 | 68.60 | 20.03 | 60.90 |
| H50kmT12hE40Pi | 13.93 | 60.12 | 18.05 | 57.06 |