# Peer review of "Revealing the sulfur dioxide emission reductions in China by assimilating surface observations in WRF-Chem"

_Atmospheric Chemistry and Physics, 2020_

## Referee Comment (RC1) · Anonymous Referee #1 · 14 Jan 2021

This manuscript developed a new emission inversion system based on 4D-LETKF and WRF-Chem to update the SO2 emission by assimilating the ground-based hourly SO2 observations. The inverted SO2 emission over China in November 2016 is well in agreement with the "bottom-up" estimation, indicating that the newly developed emission inversion system can efficiently update the SO2 emissions based on the routine surface SO2 observations. Their investigation is interesting and valuable. The manuscript is well written and structured. I recommend publication after addressing the following concerns. Line 60: There are more recent research papers of ensemble-based assimilations to estimate the emission. Feng, S., Jiang, F*., Wang, H., Wang, H., Ju, W., Shen, Y., Zheng, Y., Wu, Z. & Ding, A

(2020). NOx Emission Changes over China during the COVID-19 Epidemic Inferred from Surface NO2 Observations.Geophysical Research Letters, 47, e2020GL090080. https://doi.org/10.1029/2020GL090080 Feng, S., Jiang, F*., Wu, Z., Wang, H., Ju, W., & Wang, H. (2020). CO Emissions Inferred From Surface CO Observations Over China in December 2013 and 2017. Journal of Geophysical Research-Atmospheres, 125(7). https://doi.org/10.1029/2019JD031808 Chu, K., Z. Peng , Z. Liu, L. Lei, X. Kou, Y. Zhang, B. Xin and J. Tian: Evaluating the impact of emissions regulations on the emissions reduction during the 2015 China Victory Day Parade with an ensemble square root filter. J. Geophys. Res.-Atmos, 2018, doi:10.1002/2017JD027631 Line 76: Other two papers are also about the inverted SO2 emissions Peng, Z., Lei, L., Liu, Z., Liu, H., Chu, K., & Kou, X. (2020). Impact of assimilating meteorological observations on source emissions estimate and chemical simulations. Geophysical Research Letters, 47, e2020GL089030. https://doi.org/10.1029/2020GL089030 Peng, Z., Lei, L., Liu, Z., Sun, J., Ding, A., Ban, J., et al. (2018). The impact of multi‐species surface chemical observation assimilation on air quality forecasts in China. Atmospheric Chemistry and Physics, 18(23), 17,387–17,404. https://doi.org/10.5194/acp‐18‐17387‐2018)

Line 142: How do you decide the locations of the super-observations? Line 143: How do you decide the assimilated and independent verification observation sites? Line 180: How does the emission model forecast the emissions $E_{(t_{(n+1)})}^f$ for 12 hours? How the temporal and spatial distribution of the ensemble spread of the emissions $E_{(t_{(n+1)})}^f$ ranged? Could you please show time series of hourly ensemble spread of the emissions $E_{(t_{(n+1)})}^f$ from 00:00 UTC 8 November to 00:00 UTC 18 November 2016 and their spatial distributions at typical time. Please discuss the forecast model first since the DA depends on the details of the forecast model. Line 181-183: Please write a bit more about the generation of the initial prior ensemble of SO2 emissions. And also a bit more about the spatial distribution of the ensemble spread of the prior emissions $E_{(t_0)}$ . Line 197-200: The SO2 concentrations are updated "by recalculation of the WRF-Chem ensemble with the optimized emissions": so the uncertainties of

the forecast SO2 concentrations could still be large. This will influence the assimilation results. Please discuss a bit more about them. Line 225-230: The spatial correlations among the grid points of the forecast emissions are not clear, so are the spatial correlations among the initial prior ensemble of SO2 emissions. Figure 3: Which data are used to obtain the averaged SO2 emissions? Could you please show the difference between the analysis and MEIC2016, or the ratio?

Line 250: Are the initial and lateral boundary chemical fields perturbed?

Line 272: Could you please show time series of hourly SO2 emissions of the prior, the forecast and the analysis of the assimilation experiments from 00:00 UTC 8 November to 00:00 UTC 18 November 2016, not only the mean spatial distribution in Figure 3. These will make the reader to understand a priori value and the adjustment SO2 emissions easily. Figure 6 and 7: I guess the SO2 concentrations are obtained from the DA experiments. But I am not sure if they are the updated results by recalculation the WRF-Chem ensemble with the optimized emissions. Could you please show the difference between the updated concentrations and the original? L391: Could you please show the diurnal variations of the inverted SO2 emissions of the DA experiments?
* * *

---

## Referee Comment (RC2) · Anonymous Referee #2 · 15 Jan 2021

General Comments:

The "top-down" emission inventories of air pollutants such as sulfur dioxide are crucial to the studies of air quality prediction and emission control policy. The authors develop an emission inversion system based on the WRF-Chem model and 4D-LETKF assimilation method. This system is tested by inverting SO2 emissions with the surface observations. It takes the advantages of considering the nonlinear sulfur chemistry by ensemble forecasts with perturbed emissions, generating the flow-dependent model errors, and localizing the observation impacts. To optimize the assimilation system, the authors also make a lot of efforts to tune the inversion system parameters. The

performances of this system are evaluated by comparing to the independently updated "bottom-up" emissions. Results show that the spatial distribution and magnitude of the SO2 reductions over China are both well revealed by this system. This emission inversion system and its application are sound, and the results are convincing. I would like to recommend accepting this study after some minor revisions.

Specific Comments:

1. In ensemble data assimilation, the inflation of background covariance or the analysis covariance is generally required to avoid filter divergence. Do you use any inflation in your assimilation system? Please clarify this.

2. P5L155: As this paper employs the 4D-LETKF method, it would be helpful to clarify the '4D' /temporal features and 'L'/ spatial localization in the formulas of this method.

3. P4L132: Do you also nudge the meteorological fields in the PBL?

4. P5L155: Does the I in formula (4) represent the identity matrix?

---

## Referee Comment (RC3) · Anonymous Referee #3 · 17 Jan 2021

The manuscript used the Four-Dimensional Local Ensemble Transform Kalman Filter (4D-LETKF) and WRF-Chem to dynamically update the SO2 emission grid by grid over China by assimilating the ground-based hourly SO2 observations. The topic is relevant and useful, and the results help reduce the uncertainty of emission inventory and improving the forecasting of SO2. I recommend this paper for publication after the following points are addressed. 1.Since the implementation of strict emission mitigation strategies in 2013, there is a large reduction of SO2. These reductions are primarily caused by the relocation and/or phased out of power plants and high-emitting industrial factories. In Fig. 6, the SO2 both with MIX and the inverted emissions were underestimated around Gansu. It is not clear that the system works well when the

prior emissions were underestimated. And if the locations of emission sources have been relocated, such as the factories or power plants are built/abandoned, does the assimilation method works well? 2.In fig. 10, FR_CM with inverted emission and H50kmT1h10Ps recalculation were similar. And the results show that the simulated SO2 with inverted emission were always less than observation for all sites. Cloud that be explained? 3.Please add a) b) c) . . . etc. in figure 5, 8 and 10. And the legend of Fig.11 NCP (red line) was an error. 4.P9L265 Please add the last access date.

---

## Author Comment (AC1) · 15 Feb 2021

**Response to the Comments of Referee #1**

**Revealing the sulfur dioxide emission reductions in China by assimilating surface observations in WRF-Chem**

Tie Dai, Yueming Cheng, Daisuke Goto, Yingruo Li, Xiao Tang, Guangyu Shi, and Teruyuki Nakajima

We would like to thank to the reviewer for giving constructive criticisms, which are very helpful in improving the quality of the manuscript. We have made minor revision based on the critical comments and suggestions of the referee. The referee's comments are reproduced (black) along with our replies (blue) and changes made to the text (red) in the revised manuscript. All the authors have read the revised manuscript and agreed with submission in its revised form.

**Anonymous Referee #1**

**Comment NO.1:** *This manuscript developed a new emission inversion system based on 4D-LETKF and WRF-Chem to update the SO$_2$ emission by assimilating the ground-based hourly SO$_2$ observations. The inverted SO$_2$ emission over China in November 2016 is well in agreement with the "bottom-up" estimation, indicating that the newly developed emission inversion system can efficiently update the SO$_2$ emissions based on the routine surface SO$_2$ observations. Their investigation is interesting and valuable. The manuscript is well written and structured. I recommend publication after addressing the following concerns.*

**Response:** We thank the referee for this very positive assessment of our manuscript.

**Comment NO.2:** *Line 60: There are more recent research papers of ensemble-based assimilations to estimate the emission. Feng, S., Jiang, F*., Wang, H., Wang, H., Ju, W., Shen, Y., Zheng, Y., Wu, Z. & Ding, A (2020). NOx Emission Changes over China during the COVID-19 Epidemic Inferred from Surface NO$_2$ Observations. Geophysical Research Letters, 47, e2020GL090080. https://doi.org/10.1029/2020GL090080 Feng, S., Jiang, F*., Wu, Z., Wang, H., Ju, W., & Wang, H. (2020). CO Emissions Inferred From Surface CO Observations Over China in December 2013 and 2017. Journal of Geophysical Research-Atmospheres, 125(7). https://doi.org/10.1029/2019JD031808 Chu, K., Z. Peng , Z. Liu, L. Lei, X. Kou, Y. Zhang, B. Xin and J. Tian: Evaluating the impact of emissions regulations on the emissions reduction*

*during the 2015 China Victory Day Parade with an ensemble square root filter. J. Geophys. Res.-Atmos, 2018, doi:10.1002/2017JD027631.*

**Response:** Accept. We have added the references in the revised manuscript.

**Changes in Manuscript:** Please refer to the revised manuscript, Page 2 Lines 61-64.

**Comment NO.3:** *Line 76: Other two papers are also about the inverted $SO_2$ emissions Peng, Z., Lei, L., Liu, Z., Liu, H., Chu, K., & Kou, X. (2020). Impact of assimilating meteorological observations on source emissions estimate and chemical simulations. Geophysical Research Letters, 47, e2020GL089030. https://doi.org/10.1029/2020GL089030 Peng, Z., Lei, L., Liu, Z., Sun, J., Ding, A., Ban, J., et al. (2018). The impact of multi-species surface chemical observation assimilation on air quality forecasts in China. Atmospheric Chemistry and Physics, 18(23), 17,387–17,404. https://doi.org/10.5194/acp-18-17387-2018)*

**Response:** Accept. We have added the references in the revised manuscript.

**Changes in Manuscript:** Please refer to the revised manuscript, Page 3 Lines 80-83.

**Comment NO.4:** *Line 142: How do you decide the locations of the super-observations?*

**Response:** The locations of the super-observations are assumed as the locations of the covered model grid cells.

**Changes in Manuscript:** Please refer to the revised manuscript, Page 5 Lines 149-150.

**Comment NO.5:** *Line 143: How do you decide the assimilated and independent verification observation sites?*

**Response:** The assimilated and independent verification observation sites are randomly decided.

**Changes in Manuscript:** Please refer to the revised manuscript, Page 5 Lines 151-152.

**Comment NO.6:** *Line 180: How does the emission model forecast the emissions $E^f_{t_{n+1}}$ for 12 hours? How the temporal and spatial distribution of the ensemble spread of the emissions $E^f_{t_{n+1}}$ ranged? Could you please show time series of hourly ensemble spread of the emissions $E^f_{t_{n+1}}$ from 00:00 UTC 8 November to 00:00 UTC 18 November 2016 and their spatial distributions at typical time. Please discuss the forecast model first since the DA depends on the details of the forecast model.*

**Response:** The optimized $SO_2$ emission ensemble $E^a_{t_n}$ has $SO_2$ emissions at 12 hourly

timeslots, which are used to calculate the first guess $SO_2$ emission ensemble $E_{t_{n+1}}^f$ in sequence for the next assimilation cycle.

Time series of the hourly ensemble spreads of the forecast $SO_2$ emissions averaged over China from 00:00 UTC 8 November to 23:00 UTC 17 November 2016 are shown in Fig. S1 in the Supplement. Spatial distributions of the ensemble spreads of the forecast $SO_2$ emissions at 00:00 UTC November 13 are shown in Fig. S2 in the Supplement.

As shown in Figs. S1 and S2 in the Supplement, the temporal and spatial distributions of the ensemble spread of the forecast emissions $E_{t_{n+1}}^f$ are significantly sensitive to the assimilation system parameters. The $SO_2$ emission inversion depends on the forecast model, therefore, sensitivity experiments for various different emission forecasts are conducted to tune the assimilation system as given in Table 1.

**Changes in Manuscript:** Please refer to the revised manuscript, Page 6 Lines 194-199.

**Comment NO.7:** *Line 181-183: Please write a bit more about the generation of the initial prior ensemble of $SO_2$ emissions. And also a bit more about the spatial distribution of the ensemble spread of the prior emissions $E_{t0}$.*

**Response:** Done. The initial prior ensemble of $SO_2$ emission is generated by perturbing the freely public available MIX Asian inventory $S$ for November 2010. For example, the $SO_2$ emission for ensemble member $i$ at a given location $(x, y)$ is calculated as $f_i (x, y)S(x, y)$, and the perturbation $f_i (x, y)$, $\{i = 1,2, ..., k\}$, follows a lognormal distribution in the k-dimensional space. The mean and the variance of the perturbations $f(x, y)$ are equal to 1 and the MIX $SO_2$ uncertainty (i.e., 35%). The horizontal perfect correlated and random uncorrelated perturbations are both created to generate the initial prior ensemble $E_{t0}$ and the associated first guess $SO_2$ emission ensemble $E_{t_{n+1}}^f$. For the horizontal perfect correlated perturbations, same random perturbation factor $f_i (x, y)$ throughout the whole domain emission grids including vertical and temporal spaces per member is applied. For the horizontal random uncorrelated perturbations, the perturbation factor $f_i (x, y)$ is generated independently in horizontal space but dependently in vertical and temporal spaces. The spatial distribution of the ensemble spread of the $E_{t0}$ with either horizontal perfect correlated or random uncorrelated perturbations has the similar pattern as the MIX Asian inventory $S$, which is generally equal to 35% multiplying

*S*.

**Comment NO.8:** *Line 197-200: The $SO_2$ concentrations are updated "by recalculation of the WRF-Chem ensemble with the optimized emissions": so the uncertainties of the forecast $SO_2$ concentrations could still be large. This will influence the assimilation results. Please discuss a bit more about them.*

**Response:** Done. Theoretically, the uncertainties of the forecast $SO_2$ concentrations by recalculation of the WRF-Chem ensemble are dependent on the optimized emissions. Lower uncertainties of the initial $SO_2$ conditions for the next assimilation cycle should be found with higher accurate optimized $SO_2$ emissions, which in turn makes the $SO_2$ emission inversion more reasonable. Sensitivity experiments for the $SO_2$ emission inversions as described in section 3 are performed to choose the best assimilation system parameters.

**Comment NO.9:** *Line 225-230: The spatial correlations among the grid points of the forecast emissions are not clear, so are the spatial correlations among the initial prior ensemble of $SO_2$ emissions. Figure 3: Which data are used to obtain the averaged $SO_2$ emissions? Could you please show the difference between the analysis and MEIC 2016, or the ratio?*

**Response:** The spatial correlation coefficients among the initial prior ensemble of $SO_2$ emissions over every two model grids are equal to one, and this makes the spatial correlations among the grids points of the forecast emissions are also equal to one.

In Figure 3, the inverted $SO_2$ emissions of each assimilation experiment are obtained by averaging the ones over the ensemble members. The spatial distributions of the mean differences of the MIX and inverted $SO_2$ emissions minus the MEIC ones are shown in Fig. S3 in the Supplement, and the spatial distributions of the mean ratios between the inverted $SO_2$ emissions and the MIX ones are shown in Fig. S4 in the Supplement.

**Comment NO.10:** *Line 250: Are the initial and lateral boundary chemical fields perturbed?*

**Response:** Since we don't know the uncertainties of the global model MOZART-4/GEOS-5, the initial and lateral boundary chemical fields are not perturbed in this study.

**Changes in Manuscript:** Please refer to the revised manuscript, Page 9 Lines 282-284.

**Comment NO.11:** *Line 272: Could you please show time series of hourly $SO_2$ emissions of the prior, the forecast and the analysis of the assimilation experiments from 00:00 UTC 8 November to 00:00 UTC 18 November 2016, not only the mean spatial distribution in Figure 3. These will make the reader to understand a priori value and the adjustment $SO_2$ emissions easily. Figure 6 and 7: I guess the $SO_2$ concentrations are obtained from the DA experiments. But I am not sure if they are the updated results by recalculation the WRF-Chem ensemble with the optimized emissions. Could you please show the difference between the updated concentrations and the original?*

**Response:** Done. The time series of the hourly $SO_2$ emissions averaged over China of the initial MIX prior, the forecast and the analysis of the assimilation experiment H50kmT1hE10Ps from 00:00 UTC 8 November to 23:00 UTC 17 November 2016 are shown in Fig. S5 in the Supplement, which illustrates the adjustment of $SO_2$ emissions with data assimilation.

The $SO_2$ concentrations in each assimilation experiment are obtained by averaging the ones over the WRF-Chem ensemble recalculations with the optimized emissions. The spatial distributions of the mean $SO_2$ concentrations simulated with the original MIX emissions and the updates of the simulated $SO_2$ concentrations with the inverted $SO_2$ emissions are shown in Fig. S6 in the Supplement.

**Changes in Manuscript:** Please refer to the revised manuscript, Page 10 Lines 310-313 and Page 11 Lines 360-365.

**Comment NO.12:** *L391: Could you please show the diurnal variations of the inverted $SO_2$ emissions of the DA experiments?*

**Response:** Done. The diurnal variations of the inverted $SO_2$ emissions over China and the NCP subregion are also shown in Fig. 11c.

**Changes in Manuscript:** Please refer to the revised manuscript, Page 14 Lines 449-450.

---

## Author Response (AR1)

**Response to the Comments of Referees**

**Revealing the sulfur dioxide emission reductions in China by assimilating surface observations in WRF-Chem**

Tie Dai, Yueming Cheng, Daisuke Goto, Yingruo Li, Xiao Tang, Guangyu Shi, and Teruyuki Nakajima

We would like to thank to the reviewers for giving constructive criticisms, which are very helpful in improving the quality of the manuscript. We have made minor revision based on the critical comments and suggestions of the referees. The referees's comments are reproduced (black) along with our replies (blue) and changes made to the text (red) in the revised manuscript. All the authors have read the revised manuscript and agreed with submission in its revised form.

**Anonymous Referee #1**

**Comment NO.1:** This manuscript developed a new emission inversion system based on 4D-LETKF and WRF-Chem to update the SO2 emission by assimilating the ground-based hourly SO2 observations. The inverted SO2 emission over China in November 2016 is well in agreement with the "bottom-up" estimation, indicating that the newly developed emission inversion system can efficiently update the SO2 emissions based on the routine surface SO2 observations. Their investigation is interesting and valuable. The manuscript is well written and structured. I recommend publication after addressing the following concerns.

Response: We thank the referee for this very positive assessment of our manuscript.

**Comment NO.2:** Line 60: There are more recent research papers of ensemble-based assimilations to estimate the emission. Feng, S., Jiang, F\*., Wang, H., Wang, H., Ju, W., Shen, Y., Zheng, Y., Wu, Z. & Ding, A (2020). NOx Emission Changes over China during the COVID-19 Epidemic Inferred from Surface NO2 Observations. Geophysical Research Letters, 47, e2020GL090080. https://doi.org/10.1029/2020GL090080 Feng, S., Jiang, F\*., Wu, Z., Wang, H., Ju, W., & Wang, H. (2020). CO Emissions Inferred From Surface CO Observations Over China in December 2013 and 2017. Journal of Geophysical Research-Atmospheres, 125(7). https://doi.org/10.1029/2019JD031808 Chu, K., Z. Peng, Z. Liu, L. Lei, X. Kou, Y. Zhang, B. Xin and J. Tian: Evaluating the impact of emissions regulations on the emissions reduction

during the 2015 China Victory Day Parade with an ensemble square root filter. J. Geophys. Res.-Atmos, 2018, doi:10.1002/2017JD027631.

Response: Accept. We have added the references in the revised manuscript.

Changes in Manuscript: Please refer to the revised manuscript, Page 2 Lines 61-64.

**Comment NO.3:** Line 76: Other two papers are also about the inverted SO2 emissions Peng, Z., Lei, L., Liu, Z., Liu, H., Chu, K., & Kou, X. (2020). Impact of assimilating meteorological observations on source emissions estimate and chemical simulations. Geophysical Research Letters, 47, e2020GL089030. https://doi.org/10.1029/2020GL089030 Peng, Z., Lei, L., Liu, Z., Sun, J., Ding, A., Ban, J., et al. (2018). The impact of multi-species surface chemical observation assimilation on air quality forecasts in China. Atmospheric Chemistry and Physics, 18(23), 17,387–17,404. https://doi.org/10.5194/acp-18-17387-2018)

**Response:** Accept. We have added the references in the revised manuscript.

Changes in Manuscript: Please refer to the revised manuscript, Page 3 Lines 80-83.

**Comment NO.4:** *Line 142: How do you decide the locations of the super-observations?*

**Response:** The locations of the super-observations are assumed as the locations of the covered model grid cells.

Changes in Manuscript: Please refer to the revised manuscript, Page 5 Lines 149-150.

**Comment NO.5:** *Line 143: How do you decide the assimilated and independent verification observation sites?*

**Response:** The assimilated and independent verification observation sites are randomly decided.

Changes in Manuscript: Please refer to the revised manuscript, Page 5 Lines 151-152.

**Comment NO.6:** Line 180: How does the emission model forecast the emissions  $E_{t_{n+1}}^{f}$  for 12 hours? How the temporal and spatial distribution of the ensemble spread of the emissions  $E_{t_{n+1}}^{f}$  ranged? Could you please show time series of hourly ensemble spread of the emissions  $E_{t_{n+1}}^{f}$  from 00:00 UTC 8 November to 00:00 UTC 18 November 2016 and their spatial distributions at typical time. Please discuss the forecast model first since the DA depends on the details of the forecast model.

**Response:** The optimized SO2 emission ensemble  $E_{t_n}^a$  has SO2 emissions at 12 hourly

timeslots, which are used to calculate the first guess SO2 emission ensemble  $E_{t_{n+1}}^{f}$  in sequence for the next assimilation cycle.

Time series of the hourly ensemble spreads of the forecast  $SO_2$  emissions averaged over China from 00:00 UTC 8 November to 23:00 UTC 17 November 2016 are shown in Fig. S1 in the Supplement. Spatial distributions of the ensemble spreads of the forecast  $SO_2$  emissions at 00:00 UTC November 13 are shown in Fig. S2 in the Supplement.

As shown in Figs. S1 and S2 in the Supplement, the temporal and spatial distributions of the ensemble spread of the forecast emissions  $E_{t_{n+1}}^{f}$  are significantly sensitive to the assimilation system parameters. The SO2 emission inversion depends on the forecast model, therefore, sensitivity experiments for various different emission forecasts are conducted to tune the assimilation system as given in Table 1.

**Changes in Manuscript: Please refer to the revised manuscript, Page 6 Lines 194-199.**

**Comment NO.7:** Line 181-183: Please write a bit more about the generation of the initial prior ensemble of  $SO_2$  emissions. And also a bit more about the spatial distribution of the ensemble spread of the prior emissions  $E_{t0}$ .

**Response:** Done. The initial prior ensemble of SO2 emission is generated by perturbing the freely public available MIX Asian inventory *S* for November 2010. For example, the SO2 emission for ensemble member *i* at a given location (x, y) is calculated as  $f_i(x, y)S(x, y)$ , and the perturbation  $f_i(x, y)$ ,  $\{i = 1, 2, ..., k\}$ , follows a lognormal distribution in the k-dimensional space. The mean and the variance of the perturbations f(x, y) are equal to 1 and the MIX SO2 uncertainty (i.e., 35%). The horizontal perfect correlated and random uncorrelated perturbations are both created to generate the initial prior ensemble  $E_{t0}$  and the associated first guess SO2 emission ensemble  $E_{t_{n+1}}^f$ . For the horizontal perfect correlated perturbations, same random perturbation factor  $f_i(x, y)$  throughout the whole domain emission grids including vertical and temporal spaces per member is applied. For the horizontal random uncorrelated perturbations, the perturbation factor  $f_i(x, y)$  is generated independently in horizontal space of the ensemble spread of the  $E_{t0}$  with either horizontal perfect correlated perturbations has the similar pattern as the MIX Asian inventory *S*, which is generally equal to 35% multiplying

Changes in Manuscript: Please refer to the revised manuscript, Page 7 Lines 199-207.

**Comment NO.8:** Line 197-200: The SO2 concentrations are updated "by recalculation of the WRF-Chem ensemble with the optimized emissions": so the uncertainties of the forecast SO2 concentrations could still be large. This will influence the assimilation results. Please discuss a bit more about them.

**Response:** Done. Theoretically, the uncertainties of the forecast  $SO_2$  concentrations by recalculation of the WRF-Chem ensemble are dependent on the optimized emissions. Lower uncertainties of the initial  $SO_2$  conditions for the next assimilation cycle should be found with higher accurate optimized  $SO_2$  emissions, which in turn makes the  $SO_2$  emission inversion more reasonable. Sensitivity experiments for the  $SO_2$  emission inversions as described in section 3 are performed to choose the best assimilation system parameters.

Changes in Manuscript: Please refer to the revised manuscript, Page 7 Lines 224-229.

**Comment NO.9:** Line 225-230: The spatial correlations among the grid points of the forecast emissions are not clear, so are the spatial correlations among the initial prior ensemble of SO2 emissions. Figure 3: Which data are used to obtain the averaged SO2 emissions? Could you please show the difference between the analysis and MEIC 2016, or the ratio?

**Response:** The spatial correlation coefficients among the initial prior ensemble of  $SO_2$  emissions over every two model grids are equal to one, and this makes the spatial correlations among the grids points of the forecast emissions are also equal to one.

In Figure 3, the inverted  $SO_2$  emissions of each assimilation experiment are obtained by averaging the ones over the ensemble members. The spatial distributions of the mean differences of the MIX and inverted  $SO_2$  emissions minus the MEIC ones are shown in Fig. S3 in the Supplement, and the spatial distributions of the mean ratios between the inverted  $SO_2$  emissions and the MIX ones are shown in Fig. S4 in the Supplement.

**Changes in Manuscript:** Please refer to the revised manuscript, Page 8 Lines 257-259 and Page 10 Lines 307-310.

**Comment NO.10:** *Line 250: Are the initial and lateral boundary chemical fields perturbed?* **Response:** Since we don't know the uncertainties of the global model MOZART-4/GEOS-5, the initial and lateral boundary chemical fields are not perturbed in this study. Changes in Manuscript: Please refer to the revised manuscript, Page 9 Lines 282-284.

**Comment NO.11:** Line 272: Could you please show time series of hourly SO2 emissions of the prior, the forecast and the analysis of the assimilation experiments from 00:00 UTC 8 November to 00:00 UTC 18 November 2016, not only the mean spatial distribution in Figure 3. These will make the reader to understand a priori value and the adjustment SO2 emissions easily. Figure 6 and 7: I guess the SO2 concentrations are obtained from the DA experiments. But I am not sure if they are the updated results by recalculation the WRF-Chem ensemble with the optimized emissions. Could you please show the difference between the updated concentrations and the original?

**Response:** Done. The time series of the hourly  $SO_2$  emissions averaged over China of the initial MIX prior, the forecast and the analysis of the assimilation experiment H50kmT1hE10Ps from 00:00 UTC 8 November to 23:00 UTC 17 November 2016 are shown in Fig. S5 in the Supplement, which illustrates the adjustment of SO2 emissions with data assimilation.

The SO2 concentrations in each assimilation experiment are obtained by averaging the ones over the WRF-Chem ensemble recalculations with the optimized emissions. The spatial distributions of the mean SO2 concentrations simulated with the original MIX emissions and the updates of the simulated SO2 concentrations with the inverted SO2 emissions are shown in Fig. S6 in the Supplement.

**Changes in Manuscript:** Please refer to the revised manuscript, Page 10 Lines 310-313 and Page 11 Lines 360-365.

**Comment NO.12:** *L391: Could you please show the diurnal variations of the inverted* SO2 *emissions of the DA experiments?*

**Response:** Done. The diurnal variations of the inverted SO2 emissions over China and the NCP subregion are also shown in Fig. 11c.

Changes in Manuscript: Please refer to the revised manuscript, Page 14 Lines 449-450.

**Anonymous Referee #2**

**Comment NO.1:** The "top-down" emission inventories of air pollutants such as sulfur dioxide are crucial to the studies of air quality prediction and emission control policy. The authors develop an emission inversion system based on the WRF-Chem model and 4D-LETKF assimilation method. This system is tested by inverting SO2 emissions with the surface observations. It takes the advantages of considering the nonlinear sulfur chemistry by ensemble forecasts with perturbed emissions, generating the flow-dependent model errors, and localizing the observation impacts. To optimize the assimilation system, the authors also make a lot of efforts to tune the inversion system parameters. The performances of this system are evaluated by comparing to the independently updated "bottom-up" emissions. Results show that the spatial distribution and magnitude of the SO2 reductions over China are both well revealed by this system. This emission inversion system and its application are sound, and the results are convincing. I would like to recommend accepting this study after some minor revisions.

**Response:** We thank the referee for this very positive evaluation.

**Comment NO.2:** In ensemble data assimilation, the inflation of background covariance or the analysis covariance is generally required to avoid filter divergence. Do you use any inflation in your assimilation system? Please clarify this.

**Response:** Yes, we use the inflation of the analysis covariance in our assimilation system. We have added the multiplicative inflation factor  $\rho$  in formula (4), and the inflation factor  $\rho$  is fixed at 1.1 to inflate the analysis covariance as same as our previous studies.

Changes in Manuscript: Please refer to the revised manuscript, Page 6 Lines 173-174.

**Comment NO.3:** *P5L155:* As this paper employs the 4D-LETKF method, it would be helpful to clarify the '4D' /temporal features and 'L'/ spatial localization in the formulas of this method. **Response:** Done. We have clarified the '4D' /temporal features and 'L'/ spatial localization in the formulas (3) and (4).

Changes in Manuscript: Please refer to the revised manuscript, Page 6 Lines 174-177.

**Comment NO.4:** *P4L132: Do you also nudge the meteorological fields in the PBL?*

Response: The meteorological fields in the Planetary Boundary layer (PBL) are not nudged.

Changes in Manuscript: Please refer to the revised manuscript, Page 5 Lines 138-139.

**Comment NO.5:** P5L155: Does the I in formula (4) represent the identity matrix?

**Response:** Yes, the *I* in formula (4) represents the identity matrix.

Changes in Manuscript: Please refer to the revised manuscript, Page 6 Line 171.

**Anonymous Referee #3**

**Comment NO.1:** The manuscript used the Four-Dimensional Local Ensemble Transform Kalman Filter (4D-LETKF) and WRF-Chem to dynamically update the SO2 emission grid by grid over China by assimilating the ground-based hourly SO2 observations. The topic is relevant and useful, and the results help reduce the uncertainty of emission inventory and improving the forecasting of SO2. I recommend this paper for publication after the following points are addressed.

**Response:** We thank the referee for this very positive assessment of our manuscript.

**Comment NO.2:** Since the implementation of strict emission mitigation strategies in 2013, there is a large reduction of SO2. These reductions are primarily caused by the relocation and/or phased out of power plants and high-emitting industrial factories. In Fig. 6, the SO2 both with MIX and the inverted emissions were underestimated around Gansu. It is not clear that the system works well when the prior emissions were underestimated. And if the locations of emission sources have been relocated, such as the factories or power plants are built/abandoned, does the assimilation method works well?

**Response:** Agree. The underestimation of the surface  $SO_2$  concentration with the original MIX emission over northwestern China such as the Gansu province is potentially attributable to the increasing  $SO_2$  emissions due to energy industry expansion and relocation over northwestern China. The  $SO_2$  emissions and surface concentrations over the Gansu province are increased to reduce the negative biases in the assimilation experiments as shown in Figs. S4 and S6 in the Supplement, indicating our emission inversion system also works well when the prior emissions are underestimated. However, the simulated surface  $SO_2$  concentrations with the inverted emissions are still underestimated over the Gansu province. The reason for the underestimation is twofold: (1) there are limited observations to be assimilated over northwestern China because the observation sites are sparse; (2) the initial priori MIX  $SO_2$  emission over northwestern China is small and underestimated, inducing the model uncertainty is small relative to the observation one. This translates to a reduced impact of the observation on the priori emission.

Changes in Manuscript: Please refer to the revised manuscript, Page 12 Lines 396-405. Comment NO.3: In fig. 10, FR\_CM with inverted emission and H50kmT1h10Ps recalculation were similar. And the results show that the simulated  $SO_2$  with inverted emission were always less than observation for all sites. Cloud that be explained?

**Response:** Yes, it could be explained. The simulated  $SO_2$  surface concentrations in all sites with the inverted emission in both the FR\_CM and assimilation recalculation are generally underestimated. This is due to the inverted emission is sufficient to reduce the overestimations of  $SO_2$  concentration over the priori  $SO_2$  emission hotspot regions but insufficient to eliminate the underestimations over northwestern China.

Changes in Manuscript: Please refer to the revised manuscript, Page 14 Lines 444-447.

**Comment NO.4:** *Please add a) b) c) : : : etc. in figure 5, 8 and 10. And the legend of Fig.11 NCP (red line) was an error.*

**Response:** Done. We have corrected the legend of Fig. 11.

Changes in Manuscript: Please refer to the revised manuscript, Figure 5, 8, 10, and 11.

Comment NO.5: P9L265 Please add the last access date.

Response: Done.

Changes in Manuscript: Please refer to the revised manuscript, Page 10 Line 297.

---

## Author Response (AR2)

**Response to the Comments of Editor**

**Revealing the sulfur dioxide emission reductions in China by assimilating surface observations in WRF-Chem**

Tie Dai, Yueming Cheng, Daisuke Goto, Yingruo Li, Xiao Tang, Guangyu Shi, and Teruyuki Nakajima

We would like to thank to the editor for giving constructive criticisms, which are very helpful in improving the quality of the manuscript. We have made minor revision based on the critical comments and suggestions of the editor. The editor's comments are reproduced (black) along with our replies (blue) and changes made to the text (red) in the revised manuscript. All the authors have read the revised manuscript and agreed with submission in its revised form.

*Comments to the Author:*

1. *Line 436, "This demonstrates" should be "It demonstrates". Some corrections are needed for some other similar sentences.*

**Response:** Done.

**Changes in Manuscript:** Please refer to the revised manuscript, Lines 58, 334, 347, and 441.

2. *Line 444, "in all sites" should be "at all sites".*

**Response:** Agree.

**Changes in Manuscript:** Please refer to the revised manuscript, Line 449.

3. *Line 457, this is grammar error in "are assumed constant".*

**Response:** Agree. We have corrected it.

**Changes in Manuscript:** Please refer to the revised manuscript, Lines 461-462.